Letter

# Schizophrenia risk conferred by rare protein-truncating variants is conserved across diverse human populations

Schizophrenia (SCZ) is a chronic mental illness and among the most debilitating conditions encountered in medical practice. A recent landmark SCZ study of the protein-coding regions of the genome identified a causal role for ten genes and a concentration of rare variant signals in evolutionarily constrained genes[1]. This recent study—and most other large-scale human genetics studies—was mainly composed of individuals of European (EUR) ancestry, and the generalizability of the findings in non-EUR populations remains unclear. To address this gap, we designed a custom sequencing panel of 161 genes selected based on the current knowledge of SCZ genetics and sequenced a new cohort of 11,580 SCZ cases and 10,555 controls of diverse ancestries. Replicating earlier work, we found that cases carried a significantly higher burden of rare protein-truncating variants (PTVs) among evolutionarily constrained genes (odds ratio = 1.48; $P = 5.4 \times 10^{-6}$). In meta-analyses with existing datasets totaling up to 35,828 cases and 107,877 controls, this excess burden was largely consistent across five ancestral populations. Two genes (*SRRM2* and *AKAP11*) were newly implicated as SCZ risk genes, and one gene (*PCLO*) was identified as shared by individuals with SCZ and those with autism. Overall, our results lend robust support to the rare allelic spectrum of the genetic architecture of SCZ being conserved across diverse human populations.

SCZ is a severe, chronic psychiatric illness associated with lifelong progression and early mortality[2–4]. The genetic architecture of SCZ includes clear contributions from common single-nucleotide polymorphisms (SNPs)[5], large copy number variants (CNVs)[6] and rare PTVs[1,7–14]. Among these, rare PTVs provide unique value by linking disease risk to individual genes unambiguously. Most recently, the Schizophrenia Exome Sequencing Meta-Analysis (SCHEMA) Consortium increased the sequenced sample size for rare PTV investigations to 24,248 SCZ cases and 97,322 controls, established the rare PTV enrichment in genes under strong evolutionary constraint and identified ten genes with excess burden of rare PTVs in cases compared with controls[1]. When considered alongside earlier studies, these results suggest that, with greater sample sizes, additional SCZ genes harboring rare PTVs will

be discovered. Whole-exome sequencing (WES) and whole-genome sequencing (WGS) remain cost prohibitive when applied at large scales, and targeted sequencing of carefully chosen genes is an alternative approach to rapidly achieve the required sample size for novel risk gene discovery.

Most large-scale human genetics research initiatives to date have failed to include diverse populations. Over 80% of genome-wide association study (GWAS) participants are of EUR ancestry, despite this group comprising less than one-quarter of the total human population[15,16]. Studies of mental illness have contributed to this disparity with almost exclusively EUR GWAS cohorts despite the roughly equal prevalence of psychiatric disorders worldwide[17]. The limited evidence from SCZ GWASs and CNV studies of non-EUR populations suggests

✉e-mail: dol31@pitt.edu; laura.huckins@mssm.edu; alexander.charney@mssm.edu

**Fig. 1 | Study design and cohort ancestry composition. a**, Overview of the study design. **b**, Gene selection for the targeted sequencing panel. Genes were selected based on a combination of previous association statistics (SCHEMA), gTADA rankings and GWAS associations. Specially, we included: (1) genes in the top 100 based on the gTADA rank and/or the SCHEMA $P$ value (top 100 in SCHEMA and gTADA, top 100 in SCHEMA alone and top 100 in gTADA alone; total $n = 133$ genes); (2) genes with evidence for association with SCZ in both GWASs and SCHEMA (special GWAS genes; $n = 4$ genes); and (3) an additional 24 genes that had the best 24 gTADA rankings of the remaining genes with a burden $P$ value

of <0.05, to fill up the target panel. The $x$ axis shows the gene-level $P$ value using SCHEMA interim data, based on which the panel was constructed (different from the final published version). The $y$ axis shows the gTADA rank of genes. Only the top 500 genes are plotted for a clear display. Some highly ranked genes were excluded (gray dots) due to logistic issues during panel construction. **c**, PGC3SEQ ancestry composition. PGC3SEQ samples include substantial non-EUR ancestry. The first two principal components (PCs) are plotted along each axis, colored by SCZ case versus control status. 1000 Genomes samples are colored by super-population.

broadly shared genetic architecture with that of EUR populations, but ancestry-specific effects, such as the major histocompatibility complex locus in EUR populations, are also present[18–24]. For rare genetic variants, findings on a broad range of complex human traits have been largely consistent across populations[25–30]. Evidence for ancestry-specific rare variant effects is limited but starting to emerge, such as *TMEM136* and serum lipid measurements in individuals of South Asian (SAS) ancestry[25]. No studies have yet shown the effect of rare PTVs in diverse ancestries for SCZ.

Here, to diversify populations in SCZ studies and achieve sufficient power to discover novel risk genes, we designed a custom sequencing panel of 161 putative SCZ genes and applied it to case–control cohorts totaling 22,135 individuals from diverse ancestries (40% non-EUR; Fig. 1 and Supplementary Table 1). This study, outlined in Fig. 1a and hereafter referred to as the Psychiatric Genomics Consortium Phase 3 Targeted Sequencing of Schizophrenia Study (PGC3SEQ), was limited to cohorts that were not part of earlier SCZ sequencing initiatives such as

SCHEMA. In constructing the sequencing panel, we used a data-driven algorithm to synthesize current knowledge of the genetic architecture of SCZ, including a preliminary version of the SCHEMA gene-level burden statistics[31,32], with the goal of enriching for genes likely to harbor excess rare PTVs in SCZ that had not reached exome-wide significance due to a lack of power. This algorithm[33,34] is a Bayesian framework that prioritizes genes by integrating gene-level burden statistics with gene membership in gene sets that have been implicated in SCZ (Fig. 1b and Supplementary Tables 2 and 3). The exonic regions of the 161 prioritized genes were sequenced on the Ion Torrent platform followed by rigorous quality control (Supplementary Figs. 1–6). Analyses comparing individuals with SCZ and controls were performed for rare PTVs (stop–gain, frameshift indels or essential splicing donor/acceptor) and deleterious missense variants (placed into tiers based on the missense badness, PolyPhen-2 and constraint (MPC) score[35] (tier 1: MPC > 3; tier 2: MPC 2–3; nondamaging: MPC < 2), and synonymous variants were analyzed as a negative control. In our primary analysis,

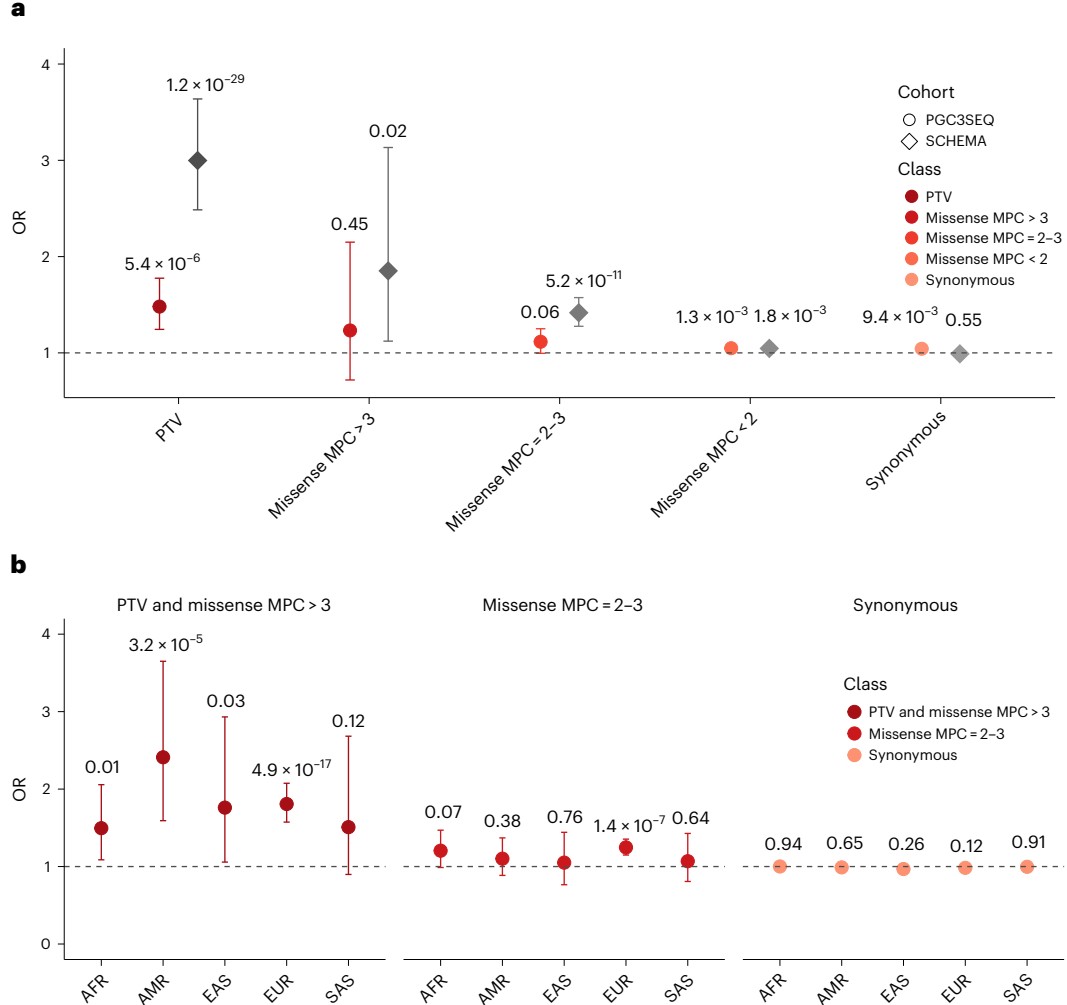

**Fig. 2 | Global enrichment in 80 panel genes under strong constraint (pLI > 0.9). a**, Case–control enrichment of rare (minor allele count ≤5) protein-truncating, missense and synonymous variants in all ancestries combined. The PGC3SEQ results were derived from 11,580 individuals with SCZ and 10,555 controls and are shown in red/orange. We conducted the same analysis in the SCHEMA samples (shown in gray; 19,108 cases and 18,001 controls) that we had access to for comparison. **b**, Ancestry-stratified rare variant (MAF < 0.1%) enrichment in the meta-analysis of PGC3SEQ and SCHEMA (29,381 cases and

27,942 controls). Three groups of variants were analyzed: PTV + MPC > 3 missense variants (combined to increase the power); MPC = 2–3 missense variants; and synonymous variants. The data are presented as point estimates of enrichment ORs (dots) and 95% confidence intervals (bars). Two-sided *P* values were calculated using Firth logistic regression, controlling for five ancestry principal components and either the rare synonymous variant count (for PTV and missense variants) or the rare nonsynonymous variant count (for synonymous variants), to control for potential unknown technical biases.

rare was defined as a minor allele count of ≤5 among the entire cohort. To maximize power, PGC3SEQ was further meta-analyzed with SCHEMA data (Supplementary Table 4 and Supplementary Fig. 7) and sequencing datasets for bipolar disorder and autism. We performed two broad types of analysis: (1) a global enrichment of all constrained genes on the custom panel (*n* = 80 genes) to investigate the overall role of rare disruptive variants in diverse ancestries; and (2) gene-level burden tests to identify novel SCZ risk genes.

PGC3SEQ SCZ cases carried a significantly higher burden of rare PTVs among the 80 constrained genes after adjusting for rare synonymous variant counts and five ancestry principal components (odds ratio (OR) = 1.48; *P* = 5.4 × 10⁻⁶; Fig. 2a and Supplementary Table 5), indicating an independent replication of the excess burden of rare PTVs observed in 3,063 constrained genes in SCHEMA. The higher effect size seen in PGC3SEQ compared with SCHEMA (OR$_{PGC3SEQ}$ = 1.48 in 80 genes; OR$_{SCHEMA}$ = 1.22 in 3,063 genes) demonstrates the effectiveness of the gene prioritization strategy used for PGC3SEQ. For the 80 genes available in both studies, the signal in PGC3SEQ was

much attenuated compared with in SCHEMA (OR$_{PGC3SEQ}$ = 1.48 versus OR$_{SCHEMA}$ = 3.0; Fig. 2a), indicating that effect sizes are probably overestimated in SCHEMA. In contrast, tier 1 and 2 missense variants were not significantly enriched in cases relative to controls in PGC3SEQ. The effects of missense variants were directionally consistent with those in SCHEMA, indicating that the insignificant results may be due to a lack of power. The burden of rare synonymous variants, which were analyzed as a negative control, was significantly higher in those with SCZ relative to controls in PGC3SEQ but not in SCHEMA. Sensitivity analysis showed that this signal was due to an overall higher burden of rare coding variants in people with SCZ relative to controls in PGC3SEQ, rather than due to technical bias or variability between contributing cohorts (Supplementary Note and Supplementary Fig. 8). The global PTV enrichment in PGC3SEQ remained significant after accounting for this overall higher baseline burden (OR = 1.4; *P* = 1.2 × 10⁻⁴; Supplementary Fig. 8c and Supplementary Table 5).

We performed meta-analyses of PGC3SEQ and SCHEMA to test whether the global enrichment signal was consistent across diverse

**Table 1 | Attempted replication of the nine significant SCHEMA genes in PGC3SEQ**

| Gene | PGC3SEQ | | | | | | SCHEMA | |
|------|---------|---|---|---|---|---|--------|---|
| | Number of PTV alleles in cases | Number of PTV alleles in controls | Number of alleles in cases | Number of alleles in controls | OR (PTVs) | Fisher's exact test *P* | OR (PTVs) | *P* |
| SETD1A | 9 | 5 | 23,160 | 21,110 | 1.64 | 0.431 | 20.1 | $2.00\times10^{-12}$ |
| CUL1 | 6 | 0 | 23,160 | 21,110 | Infinity | **0.032** | 36.1 | $2.01\times10^{-9}$ |
| XPO7 | 5 | 0 | 23,158 | 21,110 | Infinity | 0.064 | 52.2 | $7.18\times10^{-9}$ |
| TRIO | 3 | 3 | 23,160 | 21,110 | 0.91 | 1.000 | 5.0 | $6.35\times10^{-8}$ |
| CACNA1G | 6 | 13 | 23,160 | 21,110 | 0.42 | 0.105 | 3.1 | $4.57\times10^{-7}$ |
| SP4 | 1 | 0 | 23,150 | 21,104 | Infinity | 1.000 | 9.4 | $5.08\times10^{-7}$ |
| GRIN2A | 0 | 1 | 23,152 | 21,104 | 0.00 | 0.477 | 18.1 | $7.37\times10^{-7}$ |
| HERC1 | 9 | 2 | 23,160 | 21,110 | 4.10 | 0.069 | 3.5 | $1.26\times10^{-6}$ |
| RB1CC1 | 10 | 0 | 23,148 | 21,108 | Infinity | **0.002** | 10.0 | $2.00\times10^{-6}$ |
| Total | 49 | 24 | | | 1.66 | 0.027 | | |

All *P* values are two sided.

**Table 2 | Novel exome-wide significant SCZ genes**

| Gene | Location | pLI[a] | PGC3SEQ | | | SCHEMA[b] | | Meta *P*[c] | |
|------|----------|--------|---------|---|---|-----------|---|-------------|---|
| | | | Number of PTVs | OR (PTVs) | *P* | OR (PTVs) | *P* | SCZ | SCZ and ASD |
| AKAP11 | Chr13:42846289–42897396 | 0.98 | 17 | 4.26 | 0.014 | 5.25 | $8.28\times10^{-6}$ | $4.15\times10^{-7}$ | – |
| SRRM2 | Chr16:2802330–2822539 | 1 | 10 | 9.12 | 0.013 | 7.14 | $7.19\times10^{-7}$ | $7.19\times10^{-7}$ | – |
| PCLO | Chr7:82383329–82792246 | 1 | 8 | 5.01 | 0.024 | 4.02 | $9.36\times10^{-4}$ | $1.06\times10^{-5}$ | $5.84\times10^{-8}$ |

[a]Probability of loss-of-function intolerance. [b]The SCHEMA *P* values were retrieved from SCHEMA summary statistics and represent the strength of evidence from both case–control and patient–proband trio (de novo mutation) data. SCZ, meta-analysis of PGC3SEQ and SCHEMA; SCZ and ASD, SCZ further meta-analyzed with Autism Sequencing Consortium WES.
[c]Meta-analysis *P* values were determined by Stouffer's method and weighted by sample size. All *P* values are two sided.

ancestries ($n = 57,323$; ancestry breakdown in Fig. 1a). We assigned samples to five ancestral super-populations, as defined in the 1000 Genomes Project (Methods). At the aggregate level, four of the five populations displayed a higher burden of rare disruptive variants (PTV + MPC > 3 missense) in SCZ cases compared with controls ($P < 0.05$; Fig. 2b (left) and Supplementary Table 6). Although we did not find a nominally significant enrichment in the fifth ancestral population (SAS), the magnitude of enrichment was similar to that in the African (AFR) population (OR = 1.5), indicating that nonsignificance is probably a power issue (Supplementary Note and Supplementary Fig. 9). When considered separately, PGC3SEQ and SCHEMA provided independent support for the ancestry-stratified enrichments (all ancestries had OR > 1 in both datasets; Supplementary Table 6). Indeed, the PGC3SEQ data alone showed nominal significance for admixed American (AMR), East Asian (EAS) and EUR populations, exempt from any potential effect overestimation in SCHEMA. Differences, if any, in the strength of enrichment between pairs of ancestral populations were not sizable enough to be detected as significant. Across the five ancestral populations, the burden of tier 2 missense variants was evaluated, although not significant in most (OR = 1.1–1.2; Fig. 2b, middle), whereas synonymous variants were not enriched in any (Fig. 2b, right).

Having replicated the global rare PTV enrichment in PGC3SEQ and established its conservation across diverse populations, we then tested individual genes for harboring an excess burden of rare PTVs in cases relative to controls. In the PGC3SEQ data alone, none of the 161 genes sequenced were significant after Bonferroni correction ($0.05/161 = 3.1\times10^{-4}$; Supplementary Tables 7–9 and Supplementary Fig. 10). The direction of effects of all genes was overall consistent with the directions observed in SCHEMA (binomial test, $P = 0.016$) and this observation became more pronounced when considering only those 44 genes with a SCHEMA *P* value of <0.01 (binomial test, $P = 0.002$). Of the

ten significant genes identified in SCHEMA, nine were included in the PGC3SEQ panel (*GRIA3* was the lone exception). PGC3SEQ had enrichment of rare PTVs for these nine genes collectively (OR = 1.66; $P = 0.03$; 49 PTVs in cases versus 24 in controls) and two of the nine genes had $P < 0.05$ when considered individually (*RB1CC1* and *CUL1*; Table 1). Notably, *SETD1A*—the gene with the strongest signal in SCHEMA—had a nonsignificant, weakened enrichment in PGC3SEQ, suggesting an overestimation of its effect magnitude in SCHEMA ($OR_{PGC3SEQ} = 1.6$ versus $OR_{SCHEMA} = 20.1$). Another gene implicated by SCHEMA that did not find support in PGC3SEQ was *CACNA1G*. Among the nine SCHEMA genes on the PGC3SEQ panel, *CACNA1G* had the largest number of PTVs in PGC3SEQ ($n = 19$) yet an OR of 0.42, directionally inconsistent with its effect in SCHEMA ($OR_{SCHEMA} = 3.1$). Despite some evidence of winner's curse, altogether the gene-level replication tests in PGC3SEQ suggest that many of the SCHEMA genes probably confer genuine SCZ risk, including those not yet reaching exome-wide significance.

Combining PGC3SEQ and SCHEMA (totaling 35,828 cases and 107,877 controls) via a *P* value-based meta-analysis of gene-level statistics, we identified two new genes at the exome-wide significance threshold (Table 2 and Supplementary Table 7): *SRRM2* ($P = 7.2\times10^{-7}$) and *AKAP11* ($P = 4.2\times10^{-7}$). In previous work, *SRRM2* has been shown to play a role in the tauopathy of Alzheimer's disease[36–38], and de novo mutations in this gene have been linked to developmental disorders[39], while *AKAP11* was suggested as a *trans*-gene linking to a SCZ GWAS locus in a recent study[40], which, together with our results, adds to examples of the convergence of common and rare variant associations in the same gene. A recent meta-analysis of SCHEMA and a bipolar disorder dataset also found exome-wide significance for *AKAP11* (ref. [41]), suggesting a role for this gene in the shared etiology of SCZ and bipolar disorder. The current study consolidates the role of *AKAP11* in SCZ, independent of other psychiatric disorders.

Lastly, we meta-analyzed gene-level rare disruptive variant statistics from SCZ, autism spectrum disorder (ASD)[42] and bipolar disorder[41] to identify pleiotropic risk genes that are not detectable at the sample sizes attained by studies of any single disorder. This identified *PCLO* as a shared risk gene for SCZ and ASD ($P = 5.8 \times 10^{-8}$; Table 2). The result suggests that *PCLO* may be driving the common variant association at nearby loci reported in GWASs of SCZ[43] and other psychiatric disorders[44–47].

The major contribution that the PGC3SEQ study makes to the field of human genetics is demonstrating the cross-ancestry conservation of the risk conferred by a major class of genetic variation for the most severe adult mental illness. To date, the paucity of exome sequencing studies of non-EUR populations has impeded the field in developing a complete view of the genetic architecture of complex diseases, and has made it difficult to assess the degree to which rare PTV associations are susceptible to the well-known confounding effects of ancestry in GWASs and polygenic prediction studies[48–52]. Here we addressed this knowledge gap with respect to severe mental illnesses. In doing so, findings previously established in predominantly EUR cohorts have been extended to non-EUR populations for one of the major classes of genetic risk variation. This observation was not a foregone conclusion, especially since the targeted gene list was derived from SCHEMA—a study of predominantly EUR cohorts. In effect, PGC3SEQ showed that the burden signal in genes with the strongest evidence in EUR populations is conserved across non-EUR populations. Our findings are also timely information following the publication of SCHEMA, showing that some of the top genes implicated in that study are probably false positives.

There are limitations to the current study. The Ion Torrent technology is known to have decreased accuracy for indels involving homopolymer repeats of the same nucleotide[53]. We assessed the impact of such indels on our findings via a sensitivity analysis and found that excluding them would not change our conclusions (Supplementary Tables 10 and 11 and Supplementary Fig. 11). We used an interim version of the SCHEMA results for PGC3SEQ panel design, and this version is different from the published results due to changes in SCHEMA analytical strategy. Specifically, the interim SCHEMA statistics[31,32] did not include de novo mutations from trios, used a different strategy to combine PTV and missense variants and were compiled before the incorporation of Genome Aggregation Database (gnomAD) controls. Comparing the interim and published SCHEMA results, gene ranks underwent nontrivial changes, with only 27 overlapping genes between the top 100 lists in the two versions. Consequently, our panel probably included more random noise than it would have if panel construction had waited until SCHEMA was complete. As WES studies of other diseases approach the sample size achieved for SCZ, and strategies are considered for how to increase power, the current report offers valuable lessons, and we note that results on datasets as large as 24,000 cases and 50,000 controls can still change substantially as more samples are added. The possibility of such changes makes the targeted panel approach vulnerable, and perhaps WES and WGS are the safest strategies despite their cost.

In summary, rare PTVs have a robust role in SCZ, and across ancestries their effect is consistently concentrated in genes under strong evolutionary constraint. The deconvolution of this overall contribution into individual genes that may have ancestry-specific effects will require the sequencing of more individuals of diverse backgrounds. Achieving diversity in human genetic research must be a top priority to prevent health disparities from worsening as findings from genetic research begin to be translated into clinical practice.

## Online content

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

Dongjing Liu [1] ✉, Dara Meyer[1], Brian Fennessy [1], Claudia Feng [1,2], Esther Cheng [1], Jessica S. Johnson[3], You Jeong Park[1,4], Marysia-Kolbe Rieder[1], Steven Ascollillo [1], Agathe de Pins [1], Amanda Dobbyn[1,4], Dannielle Lebovitch[3], Emily Moya[1], Tan-Hoang Nguyen[5], Lillian Wilkins[1], Arsalan Hassan[6], Psychiatric Genomics Consortium Phase 3 Targeted Sequencing of Schizophrenia Study Team*, Katherine E. Burdick[7,8], Joseph D. Buxbaum [4], Enrico Domenici [9,10], Sophia Frangou[4,11], Annette M. Hartmann[12], Claudine Laurent-Levinson[13,14], Dheeraj Malhotra [15], Carlos N. Pato[16], Michele T. Pato[16], Kerry Ressler [8,17], Panos Roussos [1,3,4,18], Dan Rujescu[12], Celso Arango [19,20], Alessandro Bertolino[21], Giuseppe Blasi [21], Luisella Bocchio-Chiavetto[22,23], Dominique Campion[24,25], Vaughan Carr [26,27,28], Janice M. Fullerton [26,29], Massimo Gennarelli[23,30], Javier González-Peñas [19,20], Douglas F. Levinson[31], Bryan Mowry[32,33], Vishwajit L. Nimgaokar[34,35], Giulio Pergola [21], Antonio Rampino[21], Jorge A. Cervilla [36,37], Margarita Rivera[36,38], Sibylle G. Schwab [39], Dieter B. Wildenauer[40], Mark Daly[41,42,43,44], Benjamin Neale [41,42,43], Tarjinder Singh [41,42], Michael C. O'Donovan [45], Michael J. Owen [45], James T. Walters [45], Muhammad Ayub[46,47], Anil K. Malhotra[48,49,50], Todd Lencz [48,49,50], Patrick F. Sullivan[51,52], Pamela Sklar[3], Eli A. Stahl[3,53], Laura M. Huckins [3] ✉ & Alexander W. Charney [1,4] ✉

[1]Department of Genetics and Genomic Sciences, Icahn School of Medicine at Mount Sinai, New York, NY, USA. [2]Wellcome Sanger Institute, Hinxton, UK. [3]Pamela Sklar Division of Psychiatric Genomics, Department of Genetics and Genomic Sciences, Icahn School of Medicine at Mount Sinai, New York, NY, USA. [4]Department of Psychiatry, Icahn School of Medicine at Mount Sinai, New York, NY, USA. [5]Virginia Institute for Psychiatric and Behavioral Genetics, Department of Psychiatry, Virginia Commonwealth University, Richmond, VA, USA. [6]University of Peshawar, Peshawar, Pakistan. [7]Department of Psychiatry, Brigham and Women's Hospital, Boston, MA, USA. [8]Department of Psychiatry, Harvard Medical School, Boston, MA, USA. [9]Centre for Computational and Systems Biology, Fondazione The Microsoft Research – University of Trento, Rovereto, Italy. [10]Department of Cellular, Computational and Integrative Biology, University of Trento, Trento, Italy. [11]Djavad Mowafaghian Centre for Brain Health, University of British Columbia, Vancouver, British Columbia, Canada. [12]Department of Psychiatry and Psychotherapy, Medical University of Vienna, Vienna, Austria. [13]Faculté de Médecine Sorbonne Université, Groupe de Recherche Clinique n°15—Troubles Psychiatriques et Développement, Department of Child and Adolescent Psychiatry, Hôpital Universitaire de la Pitié-Salpêtrière, Paris, France. [14]Centre de Référence des Maladies Rares à Expression Psychiatrique, Department of Child and Adolescent Psychiatry, AP-HP Sorbonne Université, Hôpital Universitaire de la Pitié-Salpêtrière, Paris, France. [15]Department of Neuroscience and Rare Diseases, Roche Pharma Research and Early Development, F. Hoffmann-La Roche, Basel, Switzerland. [16]Department of Psychiatry and Behavioral Sciences, SUNY Downstate College of Medicine, New York, NY, USA. [17]Division of Depression and Anxiety Disorders, McLean Hospital, Belmont, MA, USA. [18]Mental Illness Research, Education, and Clinical Center (VISN 2 South), James J. Peters VA Medical Center, New York, NY, USA. [19]Department of Child and Adolescent Psychiatry, Institute of Psychiatry and Mental Health, Hospital General Universitario Gregorio Marañón, Instituto de Investigación Sanitaria Gregorio Marañón, Madrid, Spain. [20]Centro de Investigación Biomédica en Red de Salud Mental, Madrid, Spain. [21]Department of Translational Biomedicine and Neuroscience, University of Bari Aldo Moro, Bari, Italy. [22]Department of Theoretical and Applied Sciences, eCampus University, Novedrate, Italy. [23]Genetics Unit, IRCCS Istituto Centro San Giovanni di Dio Fatebenefratelli, Brescia, Italy. [24]INSERM U1245, Rouen, France. [25]Centre Hospitalier du Rouvray, Rouen, France. [26]Neuroscience Research Australia, Sydney, New South Wales, Australia. [27]School of Psychiatry, University of New South Wales, Sydney, New South Wales, Australia. [28]Department of Psychiatry, School of Clinical Sciences, Monash University, Melbourne, Victoria, Australia. [29]School of Medical Sciences, University of New South Wales, Sydney, New South Wales, Australia. [30]Department of Molecular and Translational Medicine, University of Brescia, Brescia, Italy. [31]Department of Psychiatry, Stanford University, Stanford, CA, USA. [32]Queensland Brain Institute, The University of Queensland, Brisbane, Queensland, Australia. [33]Queensland Centre for Mental Health Research, The University of Queensland, Brisbane, Queensland, Australia. [34]Department of Psychiatry, University of Pittsburgh School of Medicine, Western Psychiatric Hospital, Pittsburgh, PA, USA. [35]Department of Human Genetics, Graduate School of Public Health, University of Pittsburgh, Pittsburgh, PA, USA. [36]Institute of Neurosciences, Biomedical Research Centre, University of Granada, Granada, Spain. [37]Department of Psychiatry, San Cecilio University Hospital, University of Granada, Granada, Spain. [38]Department of Biochemistry and Molecular Biology II, Faculty of Pharmacy, University of Granada, Granada, Spain. [39]Molecular Horizons, Faculty of Science, Medicine and Health, University of Wollongong, Wollongong, New South Wales, Australia. [40]The University of Western Australia, Perth, Western Australia, Australia. [41]Analytic and Translational Genetics Unit, Department of Medicine, Massachusetts General Hospital, Boston, MA, USA. [42]Stanley Center for Psychiatric Research, Broad Institute of MIT and Harvard, Cambridge, MA, USA. [43]Program in Medical and Population Genetics, Broad Institute of MIT and Harvard, Cambridge, MA, USA. [44]Institute for Molecular Medicine Finland, University of Helsinki, Helsinki, Finland. [45]MRC Centre for Neuropsychiatric Genetics and Genomics, Division of Psychological Medicine and Clinical Neurosciences, Cardiff University, Cardiff, UK. [46]University College London, London, UK. [47]Department of Psychiatry, Queen's University, Kingston, Ontario, Canada. [48]Department of Psychiatry, Zucker School of Medicine at Hofstra/Northwell, Hempstead, NY, USA. [49]Institute for Behavioral Science, Feinstein Institutes for Medical Research, Manhasset, NY, USA. [50]Division of Psychiatry Research, The Zucker Hillside Hospital, Northwell Health, New York, NY, USA. [51]Departments of Genetics and Psychiatry, University of North Carolina, Chapel Hill, NC, USA. [52]Department of Medical Epidemiology and Biostatistics, Karolinska Institutet, Stockholm, Sweden. [53]Regeneron Pharmaceuticals, Tarrytown, NY, USA. *A list of authors and their affiliations appears at the end of the paper. ✉e-mail: dol31@pitt.edu; laura.huckins@mssm.edu; alexander.charney@mssm.edu

**Psychiatric Genomics Consortium Phase 3 Targeted Sequencing of Schizophrenia Study Team**

Henry S. Aghanwa[54], Moin Ansari[55], Aftab Asif[56], Rubina Aslam[57], Jose L. Ayuso[20,58,59], Tim Bigdeli[60], Stefano Bignotti[61], Julio Bobes[20,62], Bekh Bradley[63], Peter Buckley[64], Murray J. Cairns[65,66,67], Stanley V. Catts[68,69], Abdul Rashid Chaudhry[70], David Cohen[13,14,71], Brett L. Collins[4], Angèle Consoli[13,14], Javier Costas[72], Benedicto Crespo-Facorro[58,73], Nikolaos P. Daskalakis[74,75], Michael Davidson[76], Kenneth L. Davis[77], Faith Dickerson[78], Imtiaz A. Dogar[79], Elodie Drapeau[4], Lourdes Fañanás[58,80], Ayman Fanous[81,82], Warda Fatima[83], Mar Fatjo[20,84,85], Cheryl Filippich[32,33], Joseph Friedman[4], John F. Fullard[1], Penelope Georgakopoulos[16], Marianna Giannitelli[13,14], Ina Giegling[12], Melissa J. Green[26,27], Olivier Guillin[24,25,86], Blanca Gutierrez[36,87], Herlina Y. Handoko[88], Stella Kim Hansen[16], Maryam Haroon[89], Vahram Haroutunian[4], Frans A. Henskens[90], Fahad Hussain[91], Assen V. Jablensky[92], Jamil Junejo[93], Brian J. Kelly[90], Shams-ud-Din A. Khan[94], Muhammad N. S. Khan[95],

Letter

Anisuzzaman Khan[96], Hamid R. Khawaja[97], Bakht Khizar[91], Steven P. Kleopoulos[1], James Knowles[60], Bettina Konte[12], Agung A. A. A. Kusumawardhani[98], Naeemullah Leghari[99], Xudong Liu[95], Adriana Lori[63], Carmel M. Loughland[100], Khalid Mahmood[101], Saqib Mahmood[102], Dolores Malaspina[77], Danish Malik[103], Amy McNaughton[95], Patricia T. Michie[100], Vasiliki Michopolous[63], Esther Molina[36,104], María D. Molto[20,105], Asim Munir[106], Gerard Muntané[20,107], Farooq Naeem[108], Derek J. Nancarrow[109], Amina Nasar[95], Tanvir Nasr[103], Jude U. Ohaeri[110], Jurg Ott[111], Christos Pantelis[112,113,114], Sathish Periyasamy[32,33], Ana G. Pinto[20,115], Abigail Powers[63], Belén Ramos[20,116], Nusrat H. Rana[117], Mark Rapaport[118], Abraham Reichenberg[4,119], Safaa Saker-Delye[120], Ulrich Schall[66,121], Peter R. Schofield[26,29], Rodney J. Scott[65,122], Megan Shanahan[7], Cynthia Shannon Weickert[26,27,123], Calvin Sjaarda[96], Heather J. Smith[32,33], Jose Javier Suárez-Rama[72,124], Muhammad Tariq[125], Florence Thibaut[126,127], Paul A. Tooney[65,66,67], Muhammad Umar[92], Elisabet Vilella[20,108], Mark Weiser[128], Jin Qin Wu[129] & Robert Yolken[130]

[54]St Andrew's Toowoomba Hospital, Toowoomba, Queensland, Australia. [55]Sir Cowasjee Jehangir Institute of Psychiatry, Hyderabad, Pakistan. [56]King Edward Medical University, Lahore, Pakistan. [57]Allama Iqbal Medical College, Lahore, Pakistan. [58]Department of Psychiatry, Universidad Autónoma de Madrid, Madrid, Spain. [59]Hospital Universitario de La Princesa, Instituto de Investigación Sanitaria, Madrid, Spain. [60]SUNY Downstate Health Sciences University, Brooklyn, NY, USA. [61]Psychiatry Unit, IRCCS Istituto Centro San Giovanni di Dio Fatebenefratelli, Brescia, Italy. [62]Faculty of Medicine and Health Sciences – Psychiatry, Universidad de Oviedo, Institute of Health Research of Principado de Asturias, Instituto de Neurociencias del Principado de Asturias, Oviedo, Spain. [63]Department of Psychiatry and Behavioral Sciences, Emory University, Atlanta, GA, USA. [64]Virginia Commonwealth University, Richmond, VA, USA. [65]School of Biomedical Sciences and Pharmacy, University of Newcastle, Newcastle, New South Wales, Australia. [66]Hunter Medical Research Institute, Newcastle, New South Wales, Australia. [67]Centre for Brain and Mental Health Research, The University of Newcastle, Newcastle, New South Wales, Australia. [68]Brain and Mind Centre, The University of Sydney, Sydney, New South Wales, Australia. [69]School of Medicine, The University of Queensland, Brisbane, Queensland, Australia. [70]New Millat Brain Center, Sahiwal, Pakistan. [71]Institut des Systèmes Intelligents et de Robotique, CNRS UMR7222, Sorbonne Université, Campus Pierre et Marie Curie, Faculté des Sciences et Ingénierie, Paris, France. [72]Instituto de Investigación Sanitaria de Santiago de Compostela, Complexo Hospitalario Universitario de Santiago de Compostela, Servizo Galego de Saúde, Santiago de Compostela, Spain. [73]Hospital Universitario Virgen del Rocío, Department of Psychiatry, Universidad de Sevilla, Sevilla, Spain. [74]Harvard Medical School, Boston, MA, USA. [75]McLean Hospital, Belmont, MA, USA. [76]Nicosia University School of Medicine, Nicosia, Cyprus. [77]Icahn School of Medicine at Mount Sinai, New York, NY, USA. [78]Sheppard Pratt Hospital, Baltimore, MD, USA. [79]District Headquarter Hospital Failsalbad, Failsalbad, Pakistan. [80]Department of Evolutionary Biology, Ecology and Environmental Sciences, Faculty of Biology, University of Barcelona, Barcelona, Spain. [81]University of Arizona, Tuscon, AZ, USA. [82]Veterans Affairs, New York, NY, USA. [83]University of Punjab, Lahore, Pakistan. [84]FIDMAG Germanes Hospitalàries Research Foundation, Barcelona, Spain. [85]Departament de Biologia Evolutiva, Ecologia i Ciències Ambientals, Facultat de Biologia, Universitat de Barcelona, Barcelona, Spain. [86]UFR Santé, Université de Rouen Normandie, Rouen, France. [87]Department of Psychiatry, Faculty of Medicine, University of Granada, Granada, Spain. [88]Drug Discovery Group, Cell and Molecular Biology Department, Cancer Programme, QIMR Berghofer Medical Research Institute, Brisbane, Queensland, Australia. [89]Professor Dr. Haroon Rashid Clinic, Lahore, Pakistan. [90]School of Medicine and Public Health, University of Newcastle, Newcastle, New South Wales, Australia. [91]Lahore Institute of Research and Development, Lahore, Pakistan. [92]Centre for Clinical Research in Neuropsychiatry, The University of Western Australia, Perth, Western Australia, Australia. [93]Department of Psychiatry and Behavioural Sciences, Liaquat University of Medical and Health Sciences, Jamshoro, Pakistan. [94]Al-Shamas Hospital, Sargodha, Pakistan. [95]Queen's University, Kingston, Ontario, Canada. [96]Nai Zindage Psychiatric Hospital, Multan, Pakistan. [97]Azad Jammu and Kashmir Medical College, Muzaffarabad, Pakistan. [98]Department of Psychiatry, Cipto Mangunkusumo General Hospital, Universitas Indonesia, Jakarta, Indonesia. [99]Nishtar Medical University, Multan, Pakistan. [100]School of Psychology, University of Newcastle, Newcastle, New South Wales, Australia. [101]Ar-Rahma Hospital, Multan, Pakistan. [102]University of Health Sciences, Lahore, Pakistan. [103]Ameena Clinic, Gujranwala, Pakistan. [104]Department of Nursing, Faculty of Health Sciences, University of Granada, Granada, Spain. [105]Department of Genetics, University of Valencia, Valencia, Spain. [106]Wali Neuropsychiatric Centre, Faisalabad, Pakistan. [107]Hospital Universitari Institut Pere Mata, IISPV, Universitat Rovira i Virgili, Reus, Spain. [108]Center for Addiction and Mental Health, Toronto, Ontario, Canada. [109]Department of Surgery, University of Michigan, Ann Arbor, MI, USA. [110]Department of Psychological Medicine, University of Nigeria Teaching Hospital, Enugu, Nigeria. [111]Laboratory of Statistical Genetics, The Rockefeller University, New York, NY, USA. [112]Melbourne Neuropsychiatry Centre, University of Melbourne and Melbourne Health, Melbourne, Victoria, Australia. [113]The Florey Institute of Neuroscience and Mental Health, University of Melbourne, Parkville, Melbourne, Australia. [114]NorthWestern Mental Health, Melbourne, Victoria, Australia. [115]Bioaraba Health Research Institute, OSI Araba, University Hospital, University of the Basque Country, Vitoria, Spain. [116]Parc Sanitari Sant Joan de Déu, Barcelona, Spain. [117]Punjab Institute of Mental Health, Lahore, Pakistan. [118]University of Utah, Salt Lake City, UT, USA. [119]James J. Peters VA Medical Center, New York, NY, USA. [120]Généthon, Paris, France. [121]Priority Centre for Brain and Mental Health Research, The University of Newcastle, Mater Hospital, Newcastle, New South Wales, Australia. [122]Division of Molecular Medicine, NSW Health Pathology North, Newcastle, New South Wales, Australia. [123]Department of Neuroscience, SUNY Upstate Medical University, Syracuse, NY, USA. [124]Grupo de Medicina Xenómica, Universidade de Santiago de Compostela, Santiago de Compostela, Spain. [125]Shafique Psychiatric Clinic, Peshawar, Pakistan. [126]Université de Paris, Faculté de Médecine, Hôpital Cochin-Tarnier, Paris, France. [127]INSERM U1266, Institut de Psychiatrie et de Neurosciences, Paris, France. [128]Sheba Medical Center, Ramat Gan, Israel. [129]School of Life and Environmental Sciences, University of Sydney, Sydney, New South Wales, Australia. [130]Stanley Neurovirology Laboratory, Department of Pediatrics, Johns Hopkins School of Medicine, Baltimore, MD, USA.

## Methods

### Cohorts

A brief description of the individual contributing sample collection of PGC3SEQ is available in the Supplementary Note, along with the institutional review boards that approved the sample collections. To ensure compatibility with Psychiatric Genomics Consortium definitions, we define cases as those having a diagnosis of SCZ or a schizoaffective disorder. A total of 23,352 samples selected to be nonoverlapping with SCHEMA as well as other previous and ongoing sequencing efforts in the field were identified and sequenced (Supplementary Table 1). The PGC3SEQ study protocol was approved by the Icahn School of Medicine at Mount Sinai ethical review board (16-00101).

### Gene panel construction

We intended to build a panel of putative SCZ risk genes from within which the majority of new discoveries from additional WES and WGS would come. To this end, we applied both traditional burden statistics and the generalized/gene set transmission and de novo association test (gTADA) to the SCHEMA data.

**Traditional burden statistics.** For each gene in SCHEMA, the enrichment statistics of rare variants in cases compared with controls were calculated using Fisher's exact test separately for PTVs and damaging missense variants, then the two classes of variants were combined using meta-analysis to generate a gene-level $P$ value. Of note, this gene-level $P$ value is different from that in the SCHEMA publication, which used a slightly different strategy in combining PTVs and missense variants, additionally incorporated evidence from de novo mutations using trio data and included external gnomAD controls. Such analysis strategy changes in the later stage of SCHEMA have led to nontrivial changes in gene rank, which may impact the power of our panel to implicate disease genes.

**gTADA.** gTADA is a generalized Bayesian framework where de novo and rare variant case/control data are integrated with gene-level external information to identify risk genes for neuropsychiatric disorders[33,34]. We first sought to identify gene sets associated with SCZ in SCHEMA. Through curation of the literature, we identified an initial set of ~160 candidate gene sets. Next, each set was tested independently for association with SCZ in SCHEMA data using gTADA. From all of the sets tested, we identified 27 significantly enriched gene sets. We then calculated a joint enrichment $Z$ score from the marginal $Z$ scores and the gene set correlation matrix and kept the 25 gene sets with positive joint $Z$ scores (Supplementary Table 2). For each of the 25 sets retained, gene-level statistics (posterior probability of being a risk gene) were then calculated. The genes were then ranked by this metric and the mean ranking across the 25 ranks was calculated.

Combining traditional burden statistics and gTADA, genes in the top 100 based on the gTADA mean ranking across the 25 ranks or the top 100 based on the minimum ranking across the 25 ranks and/or the top 100 based on the burden test were included in the panel (Fig. 1b and Supplementary Table 3; $n = 139$ genes; six were later removed due to the logistics of designing the sequencing panel). We next included four genes with evidence for association with SCZ in both GWASs and SCHEMA, with the criteria being: gene burden test $P$ value < 0.05; gene with a top 200 rank in gTADA; and gene start and stop positions spanning an SNP associated with SCZ in GWAS or, if not, gene located in a GWAS locus with fewer than or equal to ten genes. Finally, an additional 24 genes were chosen for inclusion by taking the best 24 gTADA rankings of the remaining genes with a burden $P$ value < 0.05.

Based on the observation that gene-level rare single-nucleotide variant burden statistics have been consistent across ancestries in a wide range of diseases[18–24], our targeted panel was expected to have broad utility across ancestries, even though its construction used EUR-dominant datasets. This was further consolidated by findings from our own ancestry-stratified analysis (Fig. 2b).

### Sequencing and variant calling

Ion AmpliSeq technology is an amplicon-based enrichment method for creating sequencing libraries. We used Ion AmpliSeq Designer version 6.13 to design amplicons that cover the exons of the 161 genes defined based on the Ion hg19 reference. The mean and median percentages of covered base pairs across all exons were 97.7 and 100%, respectively. Sequencing of the PGC3SEQ samples was performed on the Ion Torrent platform at Sema4 between June 2018 and April 2019. Sequencing plates were matched with respect to ancestry and case versus control composition whenever possible. The average sequencing depth across all samples was 224×. The Sema4 sequencing facility returned to the research team BAM files with flow signal and associated quality control metrics. Single-sample calling was performed using Torrent variantCaller version 5.8.0, which is specially optimized to exploit the underlying flow signal information generated by the Ion Torrent sequencing. Sites were left aligned and normalized and multiallelic sites were split into separate lines using BCFtools version 1.9 (http://samtools.github.io/bcftools/).

### Genotype-level quality control

We interrogated the call set with respect to a variety of quality control metrics and implement procedures to ensure rigorous quality control standards. In the absence of well-established quality control procedures specifically for Ion Torrent data, we drew on the idea of GATK's variant quality score recalibration technique and developed a machine-learning genotype-level filter based on 177 quality metrics and annotation profiles, including Ion Torrent sequencing metrics such as QUAL, FMT/GQ and FMT/DP, allele-related metrics such as AF, HRUN and MLLD and coverage and allele frequency from the gnomAD database version 2 (https://gnomad.broadinstitute.org). Considering that the majority of SCHEMA data with which we meta-analyzed were generated on the Illumina platform, we calibrated our Ion Torrent targeted sequencing data using a subset of the control samples ($n = 1,347$) with available Illumina WES data. Specifically, we used XGBoost version 1.3 (ref. [54]) in Python version 3.7.3 to train the classifier in 70% of the Ion Torrent–Illumina paired data using Illumina as the ground truth. In the remaining 30% test set, the classifier achieved an area under the curve of 0.95, an accuracy of 95.3% and a false discovery rate of 4.4% for SNPs and an accuracy of 99.0% and a false discovery rate of 6.4% for indels. Applying the trained classifier to the test dataset improved the concordance between Ion Torrent and Illumina calls from 83.1 to 95.7%. We also compared our machine-learning classifier with a set of conventional hard filters and confirmed that the classifier performs unanimously better in all metrics considered (sensitivity, specificity, accuracy and false discovery rate).

Applying the machine-learning filter to the entire dataset, 83.2% of the calls were retained, and among the passed variants, 96% were SNPs and 4% were indels. Five out of 919 detected multiallelics passed the filter and were split into multiple biallelic variants. The proportion of calls that passed the filter among samples used for model training and testing ($n = 1,347$) and the remaining samples were similar (83.9 versus 83.1%, respectively). Likewise, the pass rate among sites that were covered by both Illumina WES capture and our sequencing panel (33.8% of the calls fell into these regions) and sites only covered by our panel were comparable (85.8 versus 81.8%), indicating that the machine-learning model generalized well to new samples and new genomic regions

### Sample- and site-level quality control

To identify low-quality and outlier samples, we examined per-sample sequencing quality metrics, including the number of mapped reads, average read depth across the panel, on-target rate and uniformity rate. We also examined sample-level call set characteristics, including the call rate, inbreeding coefficient, transition-to-transversion ratio at heterozygote sites, heterozygous-to-homozygous call ratio, total number of variants, number of SNPs and indels and number of

singletons. We visualized the distribution of the above quality control metrics (Supplementary Fig. 1) and identified 94 low-quality/outlier samples that met either one of the following criteria: MappedReads < 400,000; MeanDepth < 40; OnTarget < 80; Uniformity < 65; MissingCallRate > 0.3; Inbreeding_F > 0.6; Het_Hom_Ratio < 0.6; Total_SNPs < 400; and Total_Indels < 10. The number of low-quality or outlier samples was not significantly different between cases and controls (55 out of 12,045 cases were low quality or outliers and 38 out of 11,212 controls were low quality or outliers; chi-squared test, $P = 0.1878$). All of the quality control metrics distributed similarly between SCZ cases and controls (Supplementary Fig. 2).

When combining data from single-sample calls, a no call at a particular site in a particular sample was deemed as a homozygous reference genotype if the depth at that site in that sample was greater than ten and missing otherwise. Lastly, we applied the site-level filters to exclude variants with a missing rate of >10%.

## Sample relatedness

We used the population structure-adjusted relatedness estimation methods PC-AiR and PC-Relate to estimate pairwise relatedness between samples. In addition to the quality control steps performed per previous sections, we further performed linkage disequilibrium pruning on the dataset and removed indels before relatedness estimation. Considering that the conventional kinship coefficient ranges for varying degrees of relatedness may not be appropriate when the estimates are from targeted sequencing data covering only a small fraction of the genome, we derived empirical boundaries based on the clustering of sample pairs on an identity-by-descent kinship scatterplot (Supplementary Fig. 3). The unrelated and related pairs were clearly separated into two clusters with distinct patterns (unrelated pairs: lower oval-shaped cluster; related pairs: upper left). We identified 1,096 pairs of genetic relatives and retained one sample from each pair according to the following prioritization scheme: (1) the sample has fewer genetic relatives in the entire cohort; (2) patient with SCZ; (3) the sample has available genome-wide SNP data; (4) the sample has self-reported sex information; and (5) the sample has fewer missing genotypes for variants with a minor allele frequency (MAF) of <0.1%. These measures yielded a total of 22,135 unrelated individuals for downstream analysis.

## Control for population stratification

We calculated ancestry principal components for the 22,135 unrelated individuals in PLINK version 1.9 (ref. [55]) using 1,392 linkage disequilibrium-pruned common SNPs (MAF > 1%) that passed all quality control steps. Cases and controls were broadly matched on population structures (Supplementary Fig. 5a,b). The first five principal components were used in later association analysis to control for population substructure, based on the observation that: (1) the first five principal components explained 75% of the cumulative variance in the genetic variation among study participants; and (2) the ability of principal components to separate ancestral genetic backgrounds dissipated after the first five principal components (Supplementary Fig. 5c).

## Ancestry assignment

The genetic ancestry assignment of the PGC3SEQ participants was done by calculating principal components jointly with 1000 Genomes phase 3 participants ($n = 2,501$), followed by a $K$-nearest-neighbor classification using the top three principal components. We restricted the analysis to 1,372 linkage disequilibrium-pruned common SNPs (MAF > 1%) that were present in both the study dataset and the reference dataset (1000 Genomes). The reference data were first cleaned and quality controlled using PLINK by filtering for missingness per individual (<10%) and missingness per SNP (<10%) and then subsetted to the variant set that passed all of the quality control filters in the PGC3SEQ cohort. The cleaned reference and study datasets were harmonized, combined and pruned for linkage disequilibrium, then input into PLINK for principal component analysis with default settings.

$K$-nearest-neighbor classification was used for ancestry assignment of the study participants. Cross-validation determined $K = 5$ and the first three principal components could best classify participants into five super-populations (AFR, AMR, EAS, EUR and SAS). Applying the trained classification model, we assigned each study participant to the super-population that included the most of the participant's five neighbors. About half of our study participants had self-reported ancestry and ethnicity data, which were broadly consistent with their genetically inferred ancestry. There was reasonable concordance between the country of origin of the sample collection and assigned ancestries (Supplementary Fig. 6).

We then ran another round of principal component analysis for each global population separately to generate ancestry-specific principal components, identified ancestry-specific outliers on the principal component plots and removed the outliers and recalculated the principal components until no obvious outlier existed. After two rounds of recalculation, two EAS and seven SAS individuals were flagged as outliers within ancestry and were not included in the analysis in which stratification by population was performed.

## Variant annotation

We employed the same variation annotation workflow as was used in SCHEMA for ease of replication and comparison. Specially, annotation by LOFTEE (as implemented in the Variant Effect Predictor)[56] was applied to variants that passed all quality control filters, and the analysis was restricted to the canonical transcript with the most damaging annotation. The three broad types of coding variants analyzed were: (1) PTVs, defined as any mutation that introduced a stop codon, changed the frame of the open reading frame or introduced a change at a predicted splice donor or splice acceptor site; (2) missense variants, which included any single-nucleotide variant that caused an amino acid change; and (3) synonymous variants, which resulted in no amino acid change, as a negative control. Missense variants were further partitioned into groups with increasing deleteriousness based on the MPC score annotation[35]. Tier 1 missense variants had an MPC score of >3, tier 2 missense variants had an MPC score of 2–3 and an MPC score of <2 indicated nondamaging missense variants. The use of MPC as the missense classifier was based on the SCHEMA results that were compared with Combined Annotation Dependent Depletion and PolyPhen; MPC most powerfully prioritized damaging missense de novo variants in ASD and developmental delay/intellectual disability trios[1].

## Use of SCHEMA data

SCHEMA is a large multisite collaboration aggregating, generating and analyzing high-throughput exome sequencing data of individuals with SCZ and controls to advance gene discovery. We accessed the post-quality control data of a subset of SCHEMA case–control samples with appropriate sharing permissions at the time of this work and did not reperform genotype- and sample-level filtering. Specifically, the controls from gnomAD, as included in SCHEMA, were not used in the current study due to data sharing restrictions. After excluding 216 samples detected as genetic duplicates with a PGC3SEQ sample, the available SCHEMA datasets contained 19,108 cases and 18,001 controls (Supplementary Table 4). We used the genetic ancestry label for each individual determined by the SCHEMA analysis team and, within each ancestral group, calculated population-specific principal components using linkage disequilibrium-pruned SNPs with a MAF of >1%, a call rate of >95 and a Hardy–Weinberg $P$ value of $1 \times 10^{-6}$. Using a similar procedure to that used in the PGC3SEQ data analysis, we detected and removed 24 outlier samples from the EAS group. Supplementary Fig. 7 shows the ancestral composition of the SCHEMA cohort and Supplementary Table 4 displays the number of SCHEMA cases and controls used for this study by original sample collection.

## Statistical approaches for global enrichment across constrained genes

We defined rare variants as those with a minor allele count of ≤5 in the entire sample for any ancestry-combined analysis and lifted this threshold to MAF < 0.1% in ancestry-stratified analysis to preserve power. We counted the number of rare variants by annotation type observed in each participant in individual genes and added up the counts across the 80 constrained genes. The association between rare variant burden in the gene set of interest and SCZ status was tested using logistic regression with Firth's penalized likelihood method to account for sparse data[57], while adjusting for ancestry principal components and baseline rare variant burden. The first five global principal components were used in the ancestry-combined analysis and the first four principal components calculated within each ancestry were used in ancestry-stratified analysis. The baseline rare variant burden was used to control for technical and biological differences between cases and controls. To ensure a minimum correlation between the baseline burden and the burden of interest, we used the rare synonymous variant count as the baseline burden when the burden of interest was a PTV or missense variant and the rare nonsynonymous variant count as the baseline burden when the burden of interest was a synonymous variant. The significance threshold for the enrichment analysis was determined using the Bonferroni method, correcting for the five annotation classes tested (PTVs, the three missense groups and synonymous variants); that is, 0.05/5 = 0.01. P < 0.05 was used for nominal significance.

Using the available individual-level SCHEMA data, we performed global enrichment tests across the 80 constrained genes using similar approaches as in the PGC3SEQ analysis. Specifically, we used logistic regression with Firth's correction and adjusted for ancestry principal components, sex, sequencing cohort and baseline rare variant burden. The first five global principal components were used in the ancestry-combined analysis and the first four principal components calculated within each ancestry were used in ancestry-stratified analysis.

Four of the global populations (AFR, AMR, EUR and EAS) had $n > 100$ in both PGC3SEQ and SCHEMA and we used inverse variance-weighted meta-analysis to combine their odds ratios in the two cohorts (sample size by population in Fig. 1a). To balance the power reduction due to sample stratification, we relaxed the definition of rare variants to include those with a MAF of <0.1% (compared with a minor allele count of ≤5 in the ancestry-combined analysis). In the full SCHEMA cohort, missense variants with MPC > 3 had a global signal on par with PTVs[1]; therefore, we grouped these two types of variants together in our analysis of both cohorts to further increase the power. Only PGC3SEQ contributed to the analysis of the SAS population.

## Statistical approaches for gene-based tests

Gene-based tests aggregate the effects of multiple rare variants and can increase the power to detect genetic associations[58]. It is reasonable to assume that rare disruptive variants in a gene all have the same effect direction (variant alleles associated with higher risk) and under this scenario a burden test is appropriate. Considering the sparsity of the observed count data, we used Fisher's exact test to compare the burden of PTVs in cases and controls and computed two-sided P values. The total disruptive burden per gene was quantified by adding up all PTVs (or synonymous variants, as a negative control) annotated to the gene. Different from SCHEMA, we did not incorporate missense variants because they were not significantly enriched globally (Fig. 2a). We did not pursue a meta-analysis of the PTV and MPC > 3 variants because the extremely low number of MPC > 3 variants prohibited a reliable estimation of their effect magnitude, which would be used as weights in a meta-analysis. Although Fisher's exact test is not able to accommodate covariates such as ancestry principal components and baseline burden, this did not adversely affect our analysis as the $Q$–$Q$ plot showed no sign of inflation in the statistics (Supplementary Fig. 10, top row).

In the gene-level analysis of SCHEMA, case–control cohorts and trio cohorts were meta-analyzed, and rare variants found in both types of cohort were not double counted. We combined gene-level P values from PGC3SEQ and SCHEMA (summary statistics obtained from the SCHEMA publication) using signed Stouffer's method, with the sign of the Z scores being the effect direction of the PTVs and the weights of each study calculated as:

$$\frac{4}{\frac{1}{\#cases} + \frac{1}{\#controls}} + (\#trios\ in\ SCHEMA)$$

The above equation applies equal weight to the case–control data and trio data. Since only a subset of genes had de novo mutations in SCHEMA trios and the number of trios was small relative to the case–control sample size, fine-tuning weights would not meaningfully change our results. This meta-analysis totaled 35,828 SCZ cases and 107,877 controls, representing the largest SCZ sequencing dataset to date. The exome-wide significance level was determined to be 0.05/(23,321 tests performed in SCHEMA + 161 tests performed in PGC3SEQ) = $2.13 \times 10^{-6}$. As expected, the meta-analysis P values deviated substantially from the null (Supplementary Fig. 10, middle left), consistent with an enrichment of risk genes in the targeted panel. Gene-level synonymous variant P values displayed the expected null distribution (Supplementary Fig. 10 (middle right) and Supplementary Table 9), assuring that the gene-level PTV results were free from technical or methodological artifacts agnostic to variant annotation.

We then combined the two SCZ cohorts with the WES datasets of two other psychiatric diseases to identify genes shared across diagnoses. The two studies from which we obtained summary statistics were: (1) the latest release of the Autism Sequencing Consortium (ASC)[42] (and we further converted the gene-level q values to P values); and (2) the WES of bipolar disorder by Palmer et al.[41]. Meta-analysis was performed similarly as above and the same exome-wide significance threshold was also applied ($2.13 \times 10^{-6}$). We noted some degree of control overlap between these studies (for example, SCHEMA and ASC both included Swedish controls from the same collection). As the overlap between SCHEMA and ASC consists only a small fraction of the entire sample, our analysis (and the discovery of PCLO) should only be minimally affected. The controls overlapping between SCZ and bipolar disorder are expected to be greater per contributing cohort makeup, although we did not identify any new genes.

## Reporting summary

Further information on research design is available in the Nature Portfolio Reporting Summary linked to this article.

## Data availability

We describe all of the datasets in the Methods and Supplementary Information. The raw PGC3SEQ genotype and phenotype datasets are permitted to be distributed at the individual level and we have deposited the data in the database of Genotypes and Phenotypes under accession number phs003138.v1.p1. We provide the aggregated variant counts at the gene and gene set level in Supplementary Tables 1–9. SCHEMA summary-level data are available online for viewing and download (https://schema.broadinstitute.org). SCHEMA individual-level whole-exome sequence data are hosted on the controlled-access Terra platform (https://app.terra.bio/) and shared with the collaborating study groups. Requests for access to the controlled datasets are managed by data custodians of the SCHEMA Consortium and the Broad Institute and are sent to sample contributing investigators for approval. The gnomAD database can be accessed at https://gnomad.broadinstitute.org.

## Code availability

The software and code used in this study are described in the Methods. In brief, we used Torrent variantCaller version 5.8.0 to call variants from the raw sequence data. For quality control and preprocessing, we used XGBoost version 1.3 in Python version 3.7.3, BCFtools version 1.9 and PLINK version 1.9. Reanalysis of the SCHEMA cohort was performed using Hail 0.1 and 0.2 (https://hail.is/). Main analyses of the PGC3SEQ data and their meta-analysis with SCHEMA were performed using R version 3.6 with various libraries. Visualization was generated with ggplot2 version 3.3.

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

## Acknowledgements

A.W.C. is supported by the National Institute of Mental Health (NIMH; R01MH109536). L.M.H. is supported by the NIMH (R01MH118278, R01MH124839 and U01MH109536). M.C.O., M.J.O. and J.T.W. are supported by Medical Research Council Centre grant number MR/L010305/1 and program MR/P005748/1. For sample acquisition, curation and preparation, we are grateful to Leyden Delta, Magna Laboratories and their staff (M. Helthuis, J. Jansen and A. King). We also thank L. Hopkins and the core laboratory team at Cardiff University. B.N. is supported by 1R01MH124851. K.E.B. is supported by R01MH100125 and 1I01CX000995. J.D.B. is supported by P50MH066392. J.M.F. is supported by the Janette Mary O'Neil Research Fellowship and Australian National Health and Medical Research Council (NHMRC) Project Grant 1063960. P.R.S. is supported by the Australian NHMRC Program Grant 1037196 and Investigator Grant 1176716. D.B.W. and S.G.S. are supported by NHMRC grant 513861. M.R. has been funded by Instituto de Salud Carlos III projects PI18/00238 and PI18/00467 (cofunded by the European Regional Development Fund/European Social Fund A Way to Make Europe/Investing in Your Future). C.A. has received support from the Spanish Ministry of Science and Innovation Instituto de Salud Carlos III (PI19/024), cofinanced by European Regional Development Fund Funds from the European Commission, A Way of Making Europe, Centro de Investigación Biomédica en Red de Salud Mental, Madrid Regional Government (B2017/BMD-3740 AGES-CM-2) and European Union Horizon 2020 Program under the Innovative Medicines Initiative 2 Joint Undertaking (grant agreement number 115916 (Project PRISM) and grant agreement number 777394 (Project AIMS-2-TRIALS)), Fundación Familia Alonso and Fundación Alicia Koplowitz. We acknowledge the Biobanc of Parc Sanitari Sant Joan de Déu and Centro de Investigación Biomédica en Red de Salud Mental for samples and data procurement. J.G.-P. holds a Sara Borrell grant from Instituto de Salud Carlos III (CD20/00118). B.R. has received support from the Spanish Ministry of Science and Innovation Instituto de Salud Carlos III (PI18/00213 and Miguel Servet grants CPII21/00008 and MS16/00153), cofinanced by European Regional Development Fund Funds from the European Commission. M.G. and the work at IRCCS Centro San Giovanni di Dio Fatebenefratelli is supported by the Italian Ministry of Health (Ricerca Corrente). The CommonMind datasets were generated as part of the CommonMind Consortium, supported by funding from Takeda Pharmaceutical Company, F. Hoffmann-La Roche and National Institutes of Health grants R01MH085542, R01MH093725, P50MH066392, P50MH080405, R01MH097276, RO1MH-075916, P50M096891, P50MH084053S1, R37MH057881, AG02219, AG05138, MH06692, R01MH110921, R01MH109677, R01MH109897, U01MH103392 and U01MH116442, project ZIC MH002903 and contract HHSN271201300031C through the NIMH Intramural Research Program. Brain tissue for the study was obtained from the following brain bank collections: the Mount Sinai/JJ Peters VA Medical Center NIH Brain and Tissue Repository, University of Pennsylvania Alzheimer's Disease Core Center, University of Pittsburgh Brain Tissue Donation Program and NIMH Human Brain Collection Core. P.R. is supported by R01AG067025, R01AG065582, R01AG050986, R01MH110921, U01MH116442, R01MH125246, R01MH106056 and R01MH109897. This study was supported by the Australian Schizophrenia Research Bank (chief investigators: V.C., U.S., R.J.S., A.V.J., B.M., P.T.M., S.V.C., F.A.H., C.P. and C.M.L.). The Australian Schizophrenia Research Bank is supported by Neuroscience Research Australia. The SCHEMA Consortium provided quality-controlled data on independent schizophrenia cohorts. We thank the SCHEMA Consortium team for sharing these data and results.

## Author contributions

A.W.C., P.S., L.M.H., E.A.S., P.F.S. and M.C.O. conceived of and designed the study. J.D.B., K.E.B., E.D., S.F., A.M.H., C.L.-L., D.Malhotra, C.N.P., M.T.P., K.R., P.R., D.R., C.A., A.B., G.B., L.B.-C., D.C., V.C., J.M.F., M.G., J.G.-P., D.F.L., B.M., V.L.N., G.P., A.R., J.A.C., M.R., S.G.S., D.B.W., M.D., B.N., T.S., M.C.O., M.J.O., J.T.W., M.A., A.K.M., T.L., P.F.S., P.S. and PGC3SEQ Team contributed to the original recruitment of study participants and the collection of DNA samples. A.W.C., P.S., L.M.H., E.A.S, J.S.J. and Y.J.P. coordinated attainment of the samples from original collectors. A.W.C., E.A.S., J.S.J., Y.J.P., L.W., E.M., E.C., S.A., M.-K.R., D.Lebovitch and D.Meyer contributed to sample management, processing and sequencing. D.Liu, A.W.C., L.M.H., E.A.S., B.F., C.F., A.D., T.-H.N., A.d.P. and A.H. developed statistical pipelines and performed computational analyses. D.Liu, A.W.C. and L.M.H. wrote and/or edited the manuscript. All authors reviewed and approved the paper.

## Competing interests

M.C.O., M.J.O. and J.T.W. are supported by a collaborative research grant from Takeda Pharmaceutical and Akrivia Health. A.K.M. is a consultant at Genomind and InformedDNA. D.M. is a full-time employee of F. Hoffmann-La Roche. M.D. is the Scientific Founder of Maze Therapeutics. C.A. has been a consultant to or has received honoraria or grants from Acadia, Angelini, Biogen, Boehringer, Gedeon Richter, Janssen-Cilag, Lundbeck, Medscape, Minerva, Otsuka, Pfizer, Roche, Sage, Servier, Shire, Schering-Plough, Sumitomo Dainippon Pharma, Sunovion and Takeda. D.R. served as a consultant for Janssen, received honoraria from Gerot-Lannacher, Janssen and Pharmagenetix, received travel support from Angelini and Janssen and served on the advisory boards of AC Immune, Roche and Rovi. E.A.S. is an employee of Regeneron. The remaining authors declare no competing interests.

## Additional information

**Correspondence and requests for materials** should be addressed to Dongjing Liu, Laura M. Huckins or Alexander W. Charney.

Laura M. Huckins
Alexander W. Charney

# Reporting Summary

## Statistics

For all statistical analyses, confirm that the following items are present in the figure legend, table legend, main text, or Methods section.

| n/a | Confirmed | |
|---|---|---|
| ☐ | ☒ | The exact sample size (*n*) for each experimental group/condition, given as a discrete number and unit of measurement |
| ☐ | ☒ | A statement on whether measurements were taken from distinct samples or whether the same sample was measured repeatedly |
| ☐ | ☒ | The statistical test(s) used AND whether they are one- or two-sided<br>*Only common tests should be described solely by name; describe more complex techniques in the Methods section.* |
| ☐ | ☒ | A description of all covariates tested |
| ☐ | ☒ | A description of any assumptions or corrections, such as tests of normality and adjustment for multiple comparisons |
| ☐ | ☒ | A full description of the statistical parameters including central tendency (e.g. means) or other basic estimates (e.g. regression coefficient) AND variation (e.g. standard deviation) or associated estimates of uncertainty (e.g. confidence intervals) |
| ☐ | ☒ | For null hypothesis testing, the test statistic (e.g. *F*, *t*, *r*) with confidence intervals, effect sizes, degrees of freedom and *P* value noted<br>*Give P values as exact values whenever suitable.* |
| ☒ | ☐ | For Bayesian analysis, information on the choice of priors and Markov chain Monte Carlo settings |
| ☐ | ☒ | For hierarchical and complex designs, identification of the appropriate level for tests and full reporting of outcomes |
| ☒ | ☐ | Estimates of effect sizes (e.g. Cohen's *d*, Pearson's *r*), indicating how they were calculated |

*Our web collection on statistics for biologists contains articles on many of the points above.*

## Software and code

Policy information about availability of computer code

| Data collection | No software and code was used in data collection. |
|---|---|
| Data analysis | Software and code used are described throughout the Supplementary Methods. In brief, we used Torrent Variant Caller version 5.8.0 to call variants from the raw sequence data. For QC and pre-processing, we used XGBoost v1.3 in Python v3.7.3, BCFtools v1.9, and PLINK v1.9. Re-analysis of the SCHEMA cohort was performed using Hail 0.1 and 0.2 (https://hail.is/). Main analyses in the PGC3SEQ data and its meta-analysis with SCHEMA were performed using R v3.6 with various libraries. Visualization was generated with ggplot2 v3.3. |

For manuscripts utilizing custom algorithms or software that are central to the research but not yet described in published literature, software must be made available to editors and reviewers. We strongly encourage code deposition in a community repository (e.g. GitHub). See the Nature Portfolio guidelines for submitting code & software for further information.

## Data

Policy information about availability of data

All manuscripts must include a data availability statement. This statement should provide the following information, where applicable:

- Accession codes, unique identifiers, or web links for publicly available datasets
- A description of any restrictions on data availability
- For clinical datasets or third party data, please ensure that the statement adheres to our policy

We describe all datasets in Online Methods and Supplementary tables/figures. The raw PGC3SEQ genotype and phenotype datasets are permitted to be distributed at the individual level and we have deposited the data in the database of Genotypes and Phenotypes dbGaP. The accession number is phs003138.v1.p1. We provide the aggregated variant counts at the gene and the gene-set level in supplementary tables. SCHEMA summary-level data is available as an online browser for viewing and download (https://schema.broadinstitute.org). SCHEMA individual-level whole-exome sequence data are hosted on and shared with the collaborating study

# Field-specific reporting

Please select the one below that is the best fit for your research. If you are not sure, read the appropriate sections before making your selection.

☒ Life sciences  ☐ Behavioural & social sciences  ☐ Ecological, evolutionary & environmental sciences

For a reference copy of the document with all sections, see nature.com/documents/nr-reporting-summary-flat.pdf

# Life sciences study design

All studies must disclose on these points even when the disclosure is negative.

| | |
|---|---|
| Sample size | Sample size was not predetermined in this study: we intended to aggregate samples from all available schizophrenia patient and control cohorts who have not been whole-exome or whole-genome sequenced at the time of study design. The sample size is sufficient because we were able to replicate findings from a previous study which had a bigger dataset, and meta-analyzed with that bigger study to further increase sample sizes. |
| Data exclusions | We describe sample ascertainment in detail in the Online Methods. We included only cases with a clear diagnosis of schizophrenia or schizoaffective disorders, and controls without a known diagnosis of a psychiatric disorder. We additionally described the criteria for which low-quality or related samples and low-quality variants were excluded in our study (see sections on Sample and Variant QC). |
| Replication | Our main analysis integrated case-control rare variant enrichment and gene discovery. We have access to the largest-to-date whole exome sequencing datasets of schizophrenia cohorts that are independent to our samples, and this dataset and our own dataset partially replicated one another. Some results reported in the other dataset were not replicated in our study. |
| Randomization | Case and control status of samples were assigned by investigators of contributing collections. We controlled for confounding factors (sequencing artifacts and population ancestry) by adjusting for those confounders in logistic regression. |
| Blinding | Blinding was not relevant to our study, as the genotype and phenotype data is determined/defined externally and could not be influenced by the analyst or during our aggregation steps. |

# Reporting for specific materials, systems and methods

We require information from authors about some types of materials, experimental systems and methods used in many studies. Here, indicate whether each material, system or method listed is relevant to your study. If you are not sure if a list item applies to your research, read the appropriate section before selecting a response.

## Materials & experimental systems

| n/a | Involved in the study |
|---|---|
| ☒ | ☐ Antibodies |
| ☒ | ☐ Eukaryotic cell lines |
| ☒ | ☐ Palaeontology and archaeology |
| ☒ | ☐ Animals and other organisms |
| ☐ | ☒ Human research participants |
| ☒ | ☐ Clinical data |
| ☒ | ☐ Dual use research of concern |

## Methods

| n/a | Involved in the study |
|---|---|
| ☒ | ☐ ChIP-seq |
| ☒ | ☐ Flow cytometry |
| ☒ | ☐ MRI-based neuroimaging |

## Human research participants

Policy information about studies involving human research participants

| | |
|---|---|
| Population characteristics | Supplementary Table S1 and S4 described contributing cohorts along with country of origin, the number of samples sequenced, and the number of samples retained in the final analysis. For each cohort, we give described description of the original recruitment and phenotypic ascertainment in Supplementary Information. To ensure compatibility with Psychiatric Genomics Consortium (PGC) definitions, we included samples with a diagnosis of schizophrenia and schizoaffective disorders in our analysis. The final dataset included 22,135 individuals from diverse ancestries, 40% of which are non-European (see Figure 1 for the number of subjects for each group ). We do not have complete information on subjects' age and sex. |
| Recruitment | Patients were recruited originally as a part of numerous cohort studies, described in Supplementary Table S1 and Table S4. The ascertainment strategies of contributing cohorts are described in Supplementary Information. |
| Ethics oversight | The PGC3SEQ study protocol was approved by the Icahn School of Mount Sinai ethical review board (16-00101). The IRBs |

Ethics oversight

that approved individual contributing studies are given in the Supplementary Note, Detail Cohort Description. Informed consent was obtained from all participants, and the institutional human subject review and ethics committees relevant to each contributing cohort approved the research.

Note that full information on the approval of the study protocol must also be provided in the manuscript.

