## [Peer Review File · Nature Genetics]

Peer Review Information

Manuscript Title: Conservation of rare schizophrenia risk variant burden across ancestries

Corresponding author name(s): Dr Dongjing Liu

Reviewer Comments & Decisions:

Decision Letter, initial version:
--

Dear Dongjing,

Your Letter entitled "Rare schizophrenia risk variant burden is conserved in diverse human populations" has been seen by three referees, whose comments are copied below. In the light of their advice, we have decided that we cannot offer to publish your manuscript in Nature Genetics.

In particular, while the referees find your work of some interest, they raise overlapping concerns about the level of advance your findings represent over earlier work and the strength of the novel conclusions that can be drawn from the analyses. We feel that these reservations are sufficiently important as to preclude publication of this study in Nature Genetics.

I am sorry that we cannot be more positive on this occasion but hope you will find our referees' comments helpful when preparing your paper for submission elsewhere.

Sincerely,
Kyle

Kyle Vogan, PhD
Senior Editor
Nature Genetics
<https://orcid.org/0000-0001-9565-9665>

Referee expertise:

Referee #1: Genetics, psychiatric diseases

Referee #2: Genetics, psychiatric diseases

Referee #3: Genetics, neurodevelopmental disorders

Reviewers' Comments:

Reviewer #1:

Remarks to the Author:

A recent whole-exome sequencing analysis of 24,248 schizophrenia cases and 97,322 controls of European ancestry identified ultra-rare coding variants in ten putative genes (Singh et al, 2020). In the current study, Liu et al. performed a targeted gene sequencing analysis consisting of 161 putative SCZ risk genes in additional populations with diverse ancestries. They replicated the initial findings as well as represented that the rare and disruptive variants in three additional genes are enriched in SCZ cases. Although it is a straightforward study with sound methodology, it does not make an original and high-impact contribution to the existing knowledge. In addition, there are some major points that should be clarified.

Although the title is "Rare schizophrenia risk variant burden is conserved in diverse human populations", the study actually focused on rare 'protein-truncating variants (PTVs)'. The title should be consistent with the objective and methodology.

It is stated that "The burden of rare synonymous variants, which were analyzed as a negative control, was significantly higher in SCZ relative to controls". What is the rationale of usage of rare synonymous variants as a negative control in a rare variant burden study? Instead, common variants could have been used as negative control. Inclusion of synonymous variants could have been a negative control only if the objective was to test PTVs, not rare variants.

In line 181, the authors stated that "Analyses comparing SCZ to controls were limited to rare PTVs (stop-gain, frameshift indels, or essential splicing donor/acceptor) and deleterious missense (placed into tiers based on the MPC score 35 (tier 1: MPC>3; tier 2: MPC 2~3; non-damaging: MPC". The non-disruptive rare or ultra-rare variants should also be included.

Minor comments:

1- What is the rationale of adjusting for the first five principal components (PCs)? Other PCs could also have contribution. A model can be chosen using stepAIC function of R-MASS package.

2- The colors are not clear in Figure S7. Although there are 8 different ancestries in the legend, only five distinct colors are visible on the plot. Would it be possible to change the colors or use smaller dots to make different ancestries distinguishable?

3- The Bonferroni correction threshold (including how it was calculated) should be included in line 256.

Reviewer #2:

Remarks to the Author:

In this study, the authors performed a target sequencing analysis of candidate genes using large-scale

schizophrenia case-control samples from ethnically diverse populations. The analysis of the burden of rare LOF and damaging missense variants showed that these variants contribute to the risk of schizophrenia with similar effect sizes across five continental populations. In a meta-analysis with the SCHEMA study or rare variant studies for other psychiatric disorders, the authors obtained conclusive evidence of disease association for several genes (e.g., CUL1, XPO7, RB1CC1), and also identified SRRM2, AKAP11, and PCLO as new genes exome-wide significantly associated with schizophrenia or the spectrum of schizophrenia and autism.

The methods used in this study are in general valid and well sophisticated, and the manuscript succinctly describes the key findings. Meanwhile, as indicated in the comments below, some important information seems to be lacking.

The identification of genes newly surpassed the stringent significance threshold, though the number of them may not be very large, is definitely meaningful and worth reporting. Another important finding of this study, the overall burden of rare damaging variants in schizophrenia across ethnically different populations, might be a predictable and unsurprising result, given that a large proportion of very rare variants in constrained genes are expected to be recently generated. Nevertheless, it has a certain significance to prove this with real data.

Specific comments:

1. I could not find the details of the variants found in genes of interest (e.g. the top ten genes in SCHEMA and the three genes newly reached the significance threshold in this study). For example, genomic positions, reference and variant alleles, changes at the cDNA and protein levels, etc. of these variants are definitely important information.
2. Have some of the variants reported in this study been validated in a different experimental method (e.g. Sanger sequencing or Illumina target sequencing)? While I understand that the authors have done rigorous quality controls, variants in genes of interest, such as those mentioned above (comment 1), should be experimentally validated whenever possible. In particular, indel calls in homopolymer runs should be carefully examined as it is known that error rates of these variants are high in Ion Torrent data.
3. Clinical information on individuals with a specific rare variant should be very informative. For example, it is important to know if carriers of PTV in AKAP11 and PCLO have any mood and autistic symptoms, respectively. Also, several control individuals with PTV in known neuropsychiatric/neurodevelopmental disorder genes (e.g. GRIN2A and SETD1A) were identified in this study. If these are true positive (experimentally validated) PTV, it is highly meaningful to investigate if the carriers of these variants show subthreshold psychiatric or developmental symptoms.
4. Although I recognize that PTV has a larger effect size on average than damaging ($MPC > 3$) missense variants, I suppose some useful information could be drawn from the data of missense variants in PGC3SEQ in the analysis for gene discovery. In addition, I feel that there is a methodological inconsistency because MPC3 missense variants were included in the gene-based burden analysis in SCHEMA.
5. The authors observed weak but significant enrichment of rare synonymous variants in schizophrenia in PGC3SEQ. To clarify the reason for this observation, it is important to check if there are any

differences in sample and sequence quality between cases and controls. For example, it may be helpful to know the number of outlier individuals in cases and controls in Figure S1, whether there are significant differences between cases and controls in the metrics analyzed in Figure S2, and the result of rare synonymous enrichment analysis using only the data from very high-quality samples.

6. If the enrichment of rare synonymous variants remains significant after further intensive quality controls, it would be meaningful to test the possibility that this enrichment is driven by potentially functional synonymous variants such as those in near-splice site regions or other regulatory elements.

7. In the SCHEMA study, were the data of rare de novo variants identified in probands of trio cohorts (e.g., Taiwanese and Bulgarian trios) used in both the case-control and de novo variant analyses? If so, the data of a single variant were considered twice when calculating the synthetic P value, which might have led to an inflation of the significance of specific genes. Though this is not a problem of this study itself, it is recommended to clarify this point and state it as a limitation if needed.

8. The reasoning for the calculation of the weights of the case-control and trio samples in the "Statistical approaches for gene-based tests" section of the Supplementary Material is unclear. Is it appropriate to put the same weight on the data of 3,000 trios and 1,500 vs 1,500 case-controls?

9. The numbers of controls in the analysis of global enrichment of rare damaging variants and the gene-level test are different (probably the gnomAD controls were not used in the global enrichment analysis). It is recommended to clarify this in the manuscript with the reasons.

10. Page 3 of the Supplementary Material (the "Genotype-level quality control" section); classifier -> classifier

Reviewer #3:

Remarks to the Author:

Author summary: In this manuscript, Liu and colleagues amass 22,135 individuals of diverse ancestry (called PGC3SEQ), including 11,580 cases with schizophrenia and 10,555 controls, and perform targeted sequencing of candidate schizophrenia genes based on current knowledge, for identifying overall burden of rare protein-truncating variants and novel genes that are significantly enriched in schizophrenia. The authors identify a statistically significant increase burden of rare variants in the overall cohort and for a subset of cohorts from different ancestries. The increase in burden was seen for protein-truncating variants, missense variants, as well as synonymous variants. The authors also use data from the recently conducted SCHEMA study to increase statistical power and find that when samples were assigned into super populations, there is an increase in overall burden of rare variants in cases compared to controls. When ancestral cohorts within PGC3SEQ alone were considered, only nominal significance was observed for a few populations. The authors then performed gene-level enrichment analysis but find that none of the 161 genes pass statistical significance (after multiple-testing correction), and two key genes identified by SCHEMA, SETD1A and CACNA1G, did not show replication in the PGC3SEQ cohort. Combining both cohorts, the authors identify two genes SRRM2 and AKAP11 at exome-wide significance. Finally, the authors perform a gene-level meta-analysis of schizophrenia, autism, and bipolar disorder and identify truncating mutations in PCLO as conferring shared risk for all three disorders.

General comments: The manuscript is well written and easy to follow. The authors present data on one of the largest cohorts of individuals with schizophrenia, and probably the first study to include diverse populations. Hopefully, we will see more such studies on diverse populations, beyond European population-centric studies, that identify novel genes, refine, or identify new pathways associated with schizophrenia.

Specific comments: The overall strategy of increasing statistical power to find novel schizophrenia genes that were below the significance thresholds due to limited samples sizes is commendable. However, such a strategy is heavily confounded by the populations where the original genes were discovered. The 161 genes were picked from previous knowledge of genes (including SCHEMA) that were originally discovered from European populations. Using the same genes as candidates to gain statistical enrichment in (an aggregation of) diverse populations has its drawbacks, including an inability to find single genes that reach exome-wide significance (and not able to replicate previous results from EUR populations). In addition, the authors have been upfront about the limitations of their gene panel selection (selected before the SCHEMA list was final), potentially causing noise in their selection of gene sets and contributing to false negative results in their data.

The relevance of finding increased burden of rare PTVs in the PGC3SEQ cohort is not clear, as it is possible that the burden is driven by some genes in one population but by other genes in the other populations within the cohort, all of which are collapsed in the analysis. Unless genes are picked based on specific pathways or functions (e.g., synapse), collective burden analysis is not informative.

The finding of a higher burden of synonymous variants is interesting. The authors attribute this to the overall increase in rare variant burden in cases compared to controls. This could also possibly be due to the heterogeneous nature of populations (cryptic populations substructures) some of which has been explored by the authors that is missed in the lumping of cohorts, or that there is real biology behind this enrichment (as has been noted in other studies such as Dhindsa et al, AJHG, 2020).

As stated above, inclusion of populations of diverse ancestry is a novelty. However, the reduced sample sizes make it difficult to identify enrichment for specific genes in each cohort. A trio-based analysis, which was not performed in this study, could have identified candidate genes with de novo PTVs. That is, are there some genes that are enriched/trending (or carry de novo mutations) in AFR or EAS for future follow-up?

The lack of phenotypic details is another weakness. The largely underwhelming results could also be due to the clinically heterogeneous nature of the disorder, and a potential lack of uniformity of diagnosis of individuals in the cohorts.

**Although we cannot publish your paper, it may be appropriate for another journal in the Nature Portfolio. If you wish to explore the journals and transfer your manuscript please use our

[redacted]

If you transfer to Nature journals or the Communications journals, you will not have to re-supply manuscript metadata and files. This link can only be used once and remains active until used.

All Nature Portfolio journals are editorially independent, and the decision on your manuscript will be taken by their editors. For more information, please see our [manuscript transfer FAQ](http://www.nature.com/authors/author_resources/transfer_manuscripts.html?WT.mc_id=EMI_NPG_1511_AUTHORTRANSF&WT.ec_id=AUTHOR) page.

Note that any decision to opt in to In Review at the original journal is not sent to the receiving journal on transfer. You can opt in to [In Review](https://www.nature.com/nature-research/for-authors/in-review) at receiving journals that support this service by choosing to modify your manuscript on transfer. In Review is available for primary research manuscript types only.

Author Rebuttal, Appeal

Introductory Statement from the Authors: We thank the Reviewers and editorial staff of *Nature Genetics* for the time and effort spent evaluating the merits of our manuscript. The comments provided by Reviewers were thoughtful and raised important points. In response to concerns raised about the limitation and significance of our work, in a revised manuscript we will put our findings in context and discuss important implications of our work that were not highlighted in our original submission. We will also address concerns raised about the strength of our finding by adding new analyses or supplementing existing ones. Below, in blue text, we have provided point-by-point responses to the comments of the Reviewers.

Reviewer #1:

Remarks to the Author:

A recent whole-exome sequencing analysis of 24,248 schizophrenia cases and 97,322 controls of European ancestry identified ultra-rare coding variants in ten putative genes (Singh et al, 2020). In the current study, Liu et al. performed a targeted gene sequencing analysis consisting of 161 putative SCZ risk genes in additional populations with diverse ancestries. They replicated the initial findings as well as represented that the rare and disruptive variants in three additional genes are enriched in SCZ cases. Although it is a straightforward study with sound methodology, it does not make an original and high-impact contribution to the existing knowledge. In addition, there are some major points that should be clarified.

Although the title is “Rare schizophrenia risk variant burden is conserved in diverse human populations”, the study actually focused on rare ‘protein-truncating variants (PTVs)’. The title should be consistent with the objective and methodology.

Response: We agree with the reviewer on this point and will modify the title. The new title will be, “Schizophrenia risk conferred by rare protein-truncating variants is conserved across diverse human populations”.

It is stated that “The burden of rare synonymous variants, which were analyzed as a negative control, was significantly higher in SCZ relative to controls”. What is the rationale of usage of rare synonymous variants as a negative control in a rare variant burden study? Instead, common variants could have been used as negative control. Inclusion of synonymous variants could have been a negative control only if the objective was to test PTVs, not rare variants.

Response: The objective was indeed to test rare PTVs, and we have revised the title to better aligns with the main finding on rare PTVs. Rare synonymous variants were used as a negative control because they are comparable to rare PTVs with respect to many variant-level characteristics that may associate with technical effects (e.g. allele frequency). Existing WES studies, such as SCHEMA and one of bipolar disorder (Palmer et al. 2021) also adopted such a practice.

1. Palmer, D. S. *et al.* Exome sequencing in bipolar disorder reveals shared risk gene *AKAP11* with schizophrenia. *medRxiv* 2021.03.09.21252930 (2021).

In line 181, the authors stated that “Analyses comparing SCZ to controls were limited to rare PTVs (stop-gain, frameshift indels, or essential splicing donor/acceptor) and deleterious missense (placed into tiers based on the MPC score 35 (tier 1: MPC>3; tier 2: MPC 2~3; non-damaging: MPC”. The non-disruptive rare or ultra-rare variants should also be included.

Response: We thank the reviewer for raising this point. We believe the confusion is due to slightly different terminologies used in genetic literature to refer to similar types of variants. For example, what we called “non-damaging: MPC<2” is likely what the reviewers was suggesting by “non-disruptive rare variants”, and what we defined as rare (MAC<=5) is likely the “ultra-rare variants” that the reviewer is referring to. In the potential resubmission, we would clarify the terminology to avoid confusions.

Minor comments:

- 1- What is the rationale of adjusting for the first five principal components (PCs)? Other PCs

could also have contribution. A model can be chosen using stepAIC function of R-MASS package.

Response: We thank the reviewer for pointing out this tool. We are open to try it out if such a process would help our study in other ways, given that our sensitivity analysis where different numbers of PC (5 to 20 with an increment of 1) were adjusted for showed that the results do not vary.

# PC	OR_PT V	CI_lower_PT V	CI_upper_PTV	P
5	1.48	1.24	1.78	5.40E-06
6	1.48	1.24	1.78	5.20E-06
7	1.48	1.24	1.78	5.20E-06
8	1.48	1.24	1.78	5.30E-06
9	1.47	1.24	1.77	7.50E-06
10	1.47	1.24	1.76	7.80E-06
11	1.47	1.24	1.77	7.40E-06
12	1.47	1.24	1.77	6.90E-06
13	1.47	1.24	1.77	7.00E-06
14	1.48	1.24	1.77	6.40E-06
15	1.48	1.24	1.77	6.80E-06
16	1.48	1.24	1.77	6.50E-06
17	1.48	1.24	1.77	6.50E-06
18	1.48	1.24	1.77	6.50E-06
19	1.48	1.24	1.77	6.50E-06
20	1.48	1.24	1.77	6.70E-06

The number of PCs were selected based on (1) cumulative variance explained (first five captured 74%), (2) separation of individuals with different genetically inferred ancestry on pairwise PC plots

and a parallel coordinate plot. See below for the parallel coordinate plot colored by ancestry (top) and by case-control status (bottom). The first five PCs are sufficient to represent the global population structure and can also capture some finer-scale structures.

Figure. Parallel coordinate plot of PC1-PC10, colored by genetically inferred ancestry

Figure. Parallel coordinate plot of PC1-PC10, colored by case control status

2- The colors are not clear in Figure S7. Although there are 8 different ancestries in the legend, only five distinct colors are visible on the plot. Would it be possible to change the colors or use smaller dots to make different ancestries distinguishable?

Response: Thank you for this suggestion and we have generated a new figure:

3- The Bonferroni correction threshold (including how it was calculated) should be included in line 256.

Response: We will modify line 256 to: “In the PGC3SEQ data alone, none of the 161 genes sequenced were significant after Bonferroni correction ($0.05/161=3.1\times 10^{-4}$)”

Reviewer #2:

Remarks to the Author:

In this study, the authors performed a target sequencing analysis of candidate genes using large-scale schizophrenia case-control samples from ethnically diverse populations. The analysis of the burden of rare LOF and damaging missense variants showed that these variants contribute to the risk of schizophrenia with similar effect sizes across five continental populations. In a meta-analysis with the SCHEMA study or rare variant studies for other psychiatric disorders, the authors obtained conclusive evidence of disease association for several genes (e.g., CUL1, XPO7, RB1CC1), and also identified SRRM2, AKAP11, and PCLO as new genes exome-wide significantly associated with schizophrenia or the spectrum of schizophrenia and autism.

The methods used in this study are in general valid and well sophisticated, and the manuscript succinctly describes the key findings. Meanwhile, as indicated in the comments below, some important information seems to be lacking.

The identification of genes newly surpassed the stringent significance threshold, though the number of them may not be very large, is definitely meaningful and worth reporting. Another important finding of this study, the overall burden of rare damaging variants in schizophrenia across ethnically different populations, might be a predictable and unsurprising result, given that a large proportion of very rare variants in constrained genes are expected to be recently generated. Nevertheless, it has a certain significance to prove this with real data.

Response: We thank the reviewer for acknowledging the significance of this work. Existing evidence is actually mixed in whether one should expect a conservation of rare risk variant burden across ancestries. As the reviewer has pointed out, one relevant piece of evidence is that recently generated variants are generally less affected by demographic history. However, other observations may suggest an opposite expectation: (1) 96% of the PTVs underlying the consistent enrichment signal across ancestries were observed only in a single population (i.e. different variants were analyzed for each ancestry); (2) ancestry-specific genetic risk factors are present from common variant associations. For example, the strongest GWAS hit for schizophrenia in European populations (MHC locus on chr6) is not conserved in non-European populations.

The predominant mechanism by which new genetic variants are introduced into the human species is through *de novo* mutation. Rates of *de novo* mutation across ancestries appear to be consistent, though studies designed to investigate this question have to date been limited to studying *de novo* mutation rates in less than 1,500 individuals (62% of which were EUR ancestry) and excluded ancestries that make up vast proportions of the human population (e.g., SAS, EAS) (<https://www.pnas.org/doi/full/10.1073/pnas.1902766117>). However, even assuming *de novo* mutation rates are the same across ancestries, it is not a foregone conclusion that the genetic contribution of rare PTVs to a disease (e.g., SCZ) should be consistent across all ancestral human populations. Let us imagine that two founder populations are formed from the same ancestral population. Over time, it is possible that for a given gene rare PTVs may accumulate at different rates in the two populations due to forces such as genetic drift and natural selection. This remains an active area of research (<https://www.nature.com/articles/s41576-021-00376-2>).

To back up our argument with real numbers, we show in below figure the per gene rate of rare loss-of-function variants in similarly sized EUR (x axis) and AFR (y axis) samples, calculated using the BioMe WES dataset. Rare was defined as having a $MAC \leq 5$ in the corresponding population. Rates were calculated as the number of bases within a gene that were observed to have a rare PTV in the corresponding population, divided by the length of the consensus coding sequencing of that gene. From the figure, one can see that many genes do not have a good agreement between the two populations, which also holds true when only constrained genes are considered ($pLI > 0.9$).

Figure. Comparing per gene rare LOF variants rates in two populations. Each dot is a gene, and genes are colored by their pLI

Specific comments:

1. I could not find the details of the variants found in genes of interest (e.g., the top ten genes in SCHEMA and the three genes newly reached the significance threshold in this study). For example, genomic positions, reference and variant alleles, changes at the cDNA and protein levels, etc. of these variants are definitely important information.

Response: We will provide such information as a supplementary table in a revised manuscript.

2. Have some of the variants reported in this study been validated in a different experimental method (e.g. Sanger sequencing or Illumina target sequencing)? While I understand that the authors have done rigorous quality controls, variants in genes of interest, such as those mentioned above (comment 1), should be experimentally validated whenever possible. In particular, indel calls in homopolymer runs should be carefully examined as it is known that error rates of these variants are high in Ion Torrent data.

Response: While the lack of such validation is a limitation, we believe by posing this point as a minor comment, the reviewer is on board with us that such experiments would be unlikely to change our conclusion. We were aware of the high error rate of homopolymer runs on Ion Torrent and would be happy to provide detailed information on how we closely inspected indel quality in a potential resubmission. Above all, the elevated PTV burden in SCZ cases was not driven by Indels:

Variant	OR	CI_lower	CI_upper	P
PTV	1.48	1.24	1.78	5.40E-06
PTV excluding Indels	1.81	1.35	2.47	7.50E-05

Two pieces of data showing indel calls are of high quality in our dataset: (1) the performance of our machine-learning filter was comparable between SNPs and indels (see table below); (2) among the 1,347 samples that were sequenced on both Ion-torrent and Illumina, the filtered indel calls had a consistency of 93.6% across the two platforms.

Table. Performance of the machine-learning filter on SNPs and indels

	Accuracy	Sensitivity	False discovery rate
SNP	95.3%	99.5%	4.4%
Indels	99.0%	95.3%	6.4%

3. Clinical information on individuals with a specific rare variant should be very informative. For example, it is important to know if carriers of PTV in AKAP11 and PCLO have any mood and autistic symptoms, respectively. Also, several control individuals with PTV in known neuropsychiatric/neurodevelopmental disorder genes (e.g. GRIN2A and SETD1A) were identified in this study. If these are true positive (experimentally validated) PTV, it is highly meaningful to investigate if the carriers of these variants show subthreshold psychiatric or developmental symptoms.

Response: We very much agree with the reviewer on the value of clinical data. A primary limitation of the field of psychiatric genetics over the past 15 years has been the sacrifice of phenotypic depth in order to reach to required sample sizes needed to reach sufficient power for risk variant discovery. In line with this trend, detailed phenotype data are unavailable for the vast majority of PGC3SEQ patients. A separate study to examine the prevalence of mutations on these

genes in the general population and the phenome-wide presentation of carrying such mutations would be highly informative. Ongoing initiatives are underway (led by our group and others) that are assembling sequence and clinical data from large cohorts to further elucidate the genotype-phenotype relationships.

4. Although I recognize that PTV has a larger effect size on average than damaging (MPC>3) missense variants, I suppose some useful information could be drawn from the data of missense variants in PGC3SEQ in the analysis for gene discovery. In addition, I feel that there is a methodological inconsistency because MPC3 missense variants were included in the gene-based burden analysis in SCHEMA.

Response: We thank the reviewer for bringing up this point. The decision to not include MPC3 variants was based on the observation that in combination they were not significantly enriched, and the point estimate of their OR is smaller than that of PTVs in both PGC3SEQ and SCHEMA (Figure 2A in the original submission). As a result, directly adding the count of MPC3 to the count of PTVs would decrease power of the gene-based burden tests. The alternative is to account for the differences in the effect magnitude by giving different weights to MPC3 and PTVs, but an accurate estimation of the effect magnitude of MPC3 was challenging due to their low number. Our goal was to apply the most appropriate analytic strategy for the current data, even if that means we need to customize our analysis to be slightly different from SCHEMA. In fact, the above reasoning to determine if MPC3 should be included or not follows the same as in SCHEMA.

5. The authors observed weak but significant enrichment of rare synonymous variants in schizophrenia in PGC3SEQ. To clarify the reason for this observation, it is important to check if there are any differences in sample and sequence quality between cases and controls. For example, it may be helpful to know the number of outlier individuals in cases and controls in Figure S1, whether there are significant differences between cases and controls in the metrics analyzed in Figure S2, and the result of rare synonymous enrichment analysis using only the data from very high-quality samples.

Response: We thank the reviewer for these suggestions. We agree with the reviewer that close inspection is needed to understand the underlying causes of this signal. Specifically, we must rule out any technical and methodological explanations in order to ensure the validity of the significant PTV enrichment. We described a series of inspections towards this end in the original supplementary file, under “Scrutinizing the global enrichment signal of synonymous variants”. In short, our conclusion is that the enrichment signal of PTV is robust. Per the reviewer’s suggestion, we would like now supplement the existing Figure S1 and S2 to further demonstrate that cases and controls were not meaningfully different with respect to variant-level and sample-level quality metrics.

Related to Figure S1: The number of low-quality or outlier samples was not significantly different between cases and controls (55 bad out of 12045 cases, 38 bad out of 11212, chi-square test p-value = 0.1878).

Related to Figure S2: below, we additionally provide the numbers underlying each of the boxplot. The mean and median for each of the 11 metrics are similar between cases and controls.

	mean_ctrl	mean_case	median_ctrl	median_case
Call rate	0.990	0.990	0.993	0.993
Inbreeding f	0.085	0.091	0.109	0.112
TsTv ratio	2.343	2.342	2.337	2.337
Het-Hom ratio	1.726	1.707	1.689	1.681
Total SNP	684.866	681.471	671.000	670.000
Total Indels	28.156	28.151	28.000	28.000
Singletons	2.614	2.781	2.000	2.000
Mapped reads in million	1.500	1.493	1.433	1.434
Average depth	225.442	224.421	215.600	215.300
On-target rate %	99.434	99.450	99.490	99.500
Uniformity %	93.617	93.364	94.460	94.350

6. If the enrichment of rare synonymous variants remains significant after further intensive quality controls, it would be meaningful to test the possibility that this enrichment is driven by potentially functional synonymous variants such as those in near-splice site regions or other regulatory elements.

Response: It is certainly a possibility that functional synonymous variants were enriched and a further analysis as the reviewer was suggesting may help identify such important variants. Though improving variant annotation per se is beyond the scope of the current study, we agree this work should be the goal of future follow-up studies.

7. In the SCHEMA study, were the data of rare de novo variants identified in probands of trio cohorts (e.g., Taiwanese and Bulgarian trios) used in both the case-control and de novo variant analyses? If so, the data of a single variant were considered twice when calculating the synthetic P value, which might have led to an inflation of the significance of specific genes. Though this is

not a problem of this study itself, it is recommended to clarify this point and state it as a limitation if needed.

Response: We appreciate the reviewer bringing up this point. In a revised manuscript, we would include a sentence clarifying that SCHEMA does not double-count such variants.

8. The reasoning for the calculation of the weights of the case-control and trio samples in the "Statistical approaches for gene-based tests" section of the Supplementary Material is unclear. Is it appropriate to put the same weight on the data of 3,000 trios and 1,500 vs 1,500 case-controls?

Response: We will make it clear that we calculated the weight of each gene as:

$$\frac{4}{\frac{1}{\#cases} + \frac{1}{\#controls}} + (\#trios)$$

A general finding from existing literature is that family-based studies are less powered compared to studies of unrelated individuals of the same sample size. However, it is unclear how weights should be distributed between the two types of data, as such quantities would depend on contexts and vary across genes. Since the number of trios (3,402) is relatively small compared to the number of case-control (35,828+107,877) in this study, we think the reviewer would agree with us that assigning unequal weights to the two types of data would only have minimum effects on our results. We would be happy to conduct a sensitivity analysis with trios excluded upon request.

9. The numbers of controls in the analysis of global enrichment of rare damaging variants and the gene-level test are different (probably the gnomAD controls were not used in the global enrichment analysis). It is recommended to clarify this in the manuscript with the reasons.

Response: This was due to the data sharing restrictions on the gnomAD controls. They were not included in the global enrichment test because individual-level genotype data was not available on these individuals.

10. Page 3 of the Supplementary Material (the "Genotype-level quality control" section); classier

-> classifier

Response: Thank you, we will correct this typo.

Reviewer #3:

Remarks to the Author:

Author summary: In this manuscript, Liu and colleagues amass 22,135 individuals of diverse ancestry (called PGC3SEQ), including 11,580 cases with schizophrenia and 10,555 controls, and perform targeted sequencing of candidate schizophrenia genes based on current knowledge, for identifying overall burden of rare protein-truncating variants and novel genes that are significantly enriched in schizophrenia. The authors identify a statistically significant increase burden of rare variants in the overall cohort and for a subset of cohorts from different ancestries. The increase in burden was seen for protein-truncating variants, missense variants, as well as synonymous variants. The authors also use data from the recently conducted SCHEMA study to increase statistical power and find that when samples were assigned into super populations, there is an increase in overall burden of rare variants in cases compared to controls. When ancestral cohorts within PGC3SEQ alone were considered, only nominal significance was observed for a few populations. The authors then performed gene-level enrichment analysis but find that none of the 161 genes pass statistical significance (after multiple-testing correction), and two key genes identified by SCHEMA, SETD1A and CACNA1G, did not show replication in the PGC3SEQ cohort. Combining both cohorts, the authors identify two genes SRRM2 and AKAP11 at exome-wide significance. Finally, the authors perform a gene-level meta-analysis of schizophrenia, autism, and bipolar disorder and identify truncating mutations in PCLO as conferring shared risk for all three disorders.

General comments: The manuscript is well written and easy to follow. The authors present data on one of the largest cohorts of individuals with schizophrenia, and probably the first study to include diverse populations. Hopefully, we will see more such studies on diverse populations, beyond European population-centric studies, that identify novel genes, refine, or identify new pathways associated with schizophrenia.

Specific comments: The overall strategy of increasing statistical power to find novel schizophrenia genes that were below the significance thresholds due to limited samples sizes is commendable. However, such a strategy is heavily confounded by the populations where the original genes were discovered. The 161 genes were picked from previous knowledge of genes (including SCHEMA)

that were originally discovered from European populations. Using the same genes as candidates to gain statistical enrichment in (an aggregation of) diverse populations has its drawbacks, including an inability to find single genes that reach exome-wide significance (and not able to replicate previous results from EUR populations). In addition, the authors have been upfront about the limitations of their gene panel selection (selected before the SCHEMA list was final), potentially causing noise in their selection of gene sets and contributing to false negative results in their data.

Response: We appreciate the reviewer's thoughtful comment. We acknowledged the limitations in the gene panel selection, including the fact that the best available evidence based on which we selected genes were derived from populations that were predominantly white. The fact that we had to rely on findings in European populations is a precise real example of how the historical Eurocentric bias would impede our field in making important discoveries. As such, it is of utmost priority to be able to break the vicious cycle in which the Eurocentric bias builds on and reinforces itself. We anticipate that we are still years away from having all the resources to generate WES/WGS data on large-scale non-European cohorts for unbiased gene discovery in the study of mental illness. These studies still cost upwards of tens of millions of dollars in sequencing costs. Before the field advances to that stage, a targeted sequencing strategy can still generate important and timely information for future studies to build upon. It might be true that we are underpowered to replicate SCHEMA genes for which the number of rare PTVs observed in our cohort was low (*SP4* and *GRIN2A*), but for others (*TRIO* and *CACNA1G*) where the power of a nominal replication would be sufficiently decent if their SCHEMA effect estimates were real, our data demonstrated clear evidence of false positive. This is timely information for research labs who are following SCHEMA genes through experimental studies. As such, despite limitations, our findings are timely and important, immediately on the heels of SCHEMA (published in Nature this week) showing some of their top findings are false positives as well as that some of their top findings extend to non-European populations.

The relevance of finding increased burden of rare PTVs in the PGC3SEQ cohort is not clear, as it is possible that the burden is driven by some genes in one population but by other genes in the other populations within the cohort, all of which are collapsed in the analysis. Unless genes are picked based on specific pathways or functions (e.g., synapse), collective burden analysis is not informative.

Response: We thank the reviewer for this comment. To address the first part of this comment, we would like to point the reviewer to Figure S8, where we showed that the increased burden of rare PTVs was consistent across sample collections and across ancestry groups. To address the question about the relevance of collective burden on constrained genes, we provide two reasons

explaining why this is informative. First, this means that genes conferring risk to SCZ were under strong natural selection in the past, which helps understand the origin and the evolution of the disorder in human history. Second, knowing where in the exome the PTV burden is concentrated is necessary for the development of individual-level rare variant risk score, especially given the fact that the PTV burden is not elevated across the entire exome (SCHEMA data).

The finding of a higher burden of synonymous variants is interesting. The authors attribute this to the overall increase in rare variant burden in cases compared to controls. This could also possibly be due to the heterogeneous nature of populations (cryptic populations substructures) some of which has been explored by the authors that is missed in the lumping of cohorts, or that there is real biology behind this enrichment (as has been noted in other studies such as Dhindsa et al, AJHG, 2020).

Response: Thank you for this comment. We described in detail how we inspected the enrichment signal of synonymous variants in the supplementary note, under the section “Scrutinizing the global enrichment signal of synonymous variants”. In short, our data is not consistent with an apparent impact of either residual population stratification or heterogeneity across cohorts (figure S8). Reviewer 2 also raised the possibility of functional synonymous variants. Please see our response to comment #6 of reviewer 2.

As stated above, inclusion of populations of diverse ancestry is a novelty. However, the reduced sample sizes make it difficult to identify enrichment for specific genes in each cohort. A trio-based analysis, which was not performed in this study, could have identified candidate genes with de novo PTVs. That is, are there some genes that are enriched/trending (or carry de novo mutations) in AFR or EAS for future follow-up?

Response: The top genes from our case-control ancestry-stratified analysis were *AKAP11* in EAS and *SRRM2* in AFR, both conferred risk for SCZ with p-values < 0.05. No *de novo* damaging variants on these two genes were observed in the 3,402 trios included in SCHEMA. Nonetheless, they are great candidates for future follow-up.

The lack of phenotypic details is another weakness. The largely underwhelming results could also be due to the clinically heterogeneous nature of the disorder, and a potential lack of uniformity of diagnosis of individuals in the cohorts.

Response: We agree with the reviewer that there is considerable clinical heterogeneity for schizophrenia, and by not having detailed clinical information on our cohorts we sacrificed power

for identifying genetic factors specific to a particular subtype. However, a “phenotypically homogeneous” design, although seems ideal, also have its own problems, with a pronounced one being how “homogeneous” should be defined in the absence of a biologically informed taxonomy of psychopathology, and the feasibility of applying any such definition at scale given the limited clinical data available for participants in large psychiatric genetic consortia currently. There is not enough information to guide striking the balance between a small uniform cohort and a large heterogeneous cohort. Based on results from SCHEMA (which used various patient ascertainment strategies), it is reasonable to hypothesize that a core rare variant burden is shared across patients with different clinical presentations when compared to healthy controls. Identifying this core burden is the first step in studying the contribution of rare genetic variants to SCZ risk, and future studies, as the reviewer notes, should aim to determine if specific clinical subtypes of the disease are driving this signal.

Decision Letter, Appeal

29th April 2022

Dear Dongjing,

Thank you for asking us to reconsider our decision on your manuscript "Rare schizophrenia risk variant burden is conserved in diverse human populations". I have discussed your appeal letter with my editorial colleagues, and we invite you to revise your manuscript along the lines that you propose for further editorial consideration and peer review.

When preparing a revision, please ensure that it fully complies with our editorial requirements for format and style; details can be found in the Guide to Authors on our website (<http://www.nature.com/ng/>).

Please also be sure that your manuscript is accompanied by a separate letter detailing the changes you have made and your response to the original reviews. At this stage we will need you to upload:

1) A copy of the manuscript in MS Word .docx format.

2) The Editorial Policy Checklist:

<https://www.nature.com/documents/nr-editorial-policy-checklist.pdf>

3) The Reporting Summary:

(Here you can read about the role of the Reporting Summary in reproducible science:

<https://www.nature.com/news/announcement-towards-greater-reproducibility-for-life-sciences-research-in-nature-1.22062>)

Please use the link below to be taken directly to the site and view and revise your manuscript:

[redacted]

With best wishes,
Kyle

Kyle Vogan, PhD
Senior Editor
Nature Genetics
<https://orcid.org/0000-0001-9565-9665>

Decision Letter, first revision:

31st May 2022

Dear Dongjing,

Your revised Letter "Schizophrenia risk conferred by rare protein-truncating variants is conserved across diverse human populations" has been seen by the original referees. You will see from their comments below that, while Reviewers #1 and #3 are satisfied with the revision, Reviewer #2 has expressed a few ongoing concerns. We remain interested in the possibility of publishing your study in Nature Genetics, but we would like to consider your response to these remaining concerns in the form of further revision before we make a final decision on publication.

To guide the scope of the revisions, the editors discuss the referee reports in detail within the team, including with the chief editor, with a view to identifying key priorities that should be addressed in revision, and sometimes overruling referee requests that are deemed beyond the scope of the current study. In this case, we ask that you carefully address Reviewer #2's concerns regarding the accuracy of indel calls at polynucleotide stretches by performing further experimental validation where feasible and/or revising claims in light of these concerns. We hope you will find this prioritized set of referee points to be useful when revising your study. Please do not hesitate to get in touch if you would like to discuss these issues further.

We therefore invite you to revise your manuscript taking into account all reviewer and editor comments. Please highlight all changes in the manuscript text file. At this stage, we will need you to upload a copy of the manuscript in MS Word .docx or similar editable format.

*2) If you have not done so already please begin to revise your manuscript so that it conforms to our Letter format instructions, available

[here](http://www.nature.com/ng/authors/article_types/index.html).

*3) Include a revised version of any required Reporting Summary:

[redacted]

We hope to receive your revised manuscript within 4-8 weeks. If you cannot send it within this time, please let us know.

Nature Genetics is committed to improving transparency in authorship. As part of our efforts in this direction, we are now requesting that all authors identified as 'corresponding author' on published papers create and link their Open Researcher and Contributor Identifier (ORCID) with their account on the Manuscript Tracking System (MTS), prior to acceptance. ORCID helps the scientific community achieve unambiguous attribution of all scholarly contributions. You can create and link your ORCID from the home page of the MTS by clicking on 'Modify my Springer Nature account'. For more information

please visit please visit www.springernature.com/orcid.

Sincerely,
Kyle

Kyle Vogan, PhD
Senior Editor
Nature Genetics
<https://orcid.org/0000-0001-9565-9665>

Referee expertise:

Referee #1: Genetics, psychiatric diseases

Referee #2: Genetics, psychiatric diseases

Referee #3: Genetics, neurodevelopmental disorders

Reviewers' Comments:

Reviewer #1:
Remarks to the Author:

The authors have addressed all the concerns properly.

Reviewer #2:
Remarks to the Author:

The authors have adequately addressed most of the concerns I have raised, while I don't exactly agree with the rationale indicating that there is a certain level of a priori possibility that the overall effect of

rare PTVs greatly varies across ethnicities (e.g., it is unsurprising that frequencies of individual rare PTVs derived from recent de novo mutations are different across populations and this fact does not directly support possible differences of the overall effect of rare PTVs across ethnicities; the example of MHC is for common polymorphisms and not for rare variants). Nevertheless, as I wrote, an analysis using real-world data from multiple ethnicities has a certain significance.

On the other hand, and perhaps more importantly, I have a specific concern on the accuracy of the indel calls below (one in case and four in controls) identified at a C stretch.

SETD1A 16 30977316 30965995 GCC GC frameshift variant NA 1 1 0

SETD1A 16 30977316 30965995 GCC G frameshift variant NA 1 0 1

SETD1A 16 30977321 30966000 CCC C frameshift variant NA 1 0 1

SETD1A 16 30977322 30966001 CC C frameshift variant NA 1 0 1

SETD1A 16 30977323 30966002 CG G frameshift variant NA 1 0 1

If these are false positives, it is not appropriate to conclude as follows:

Notably, SETD1A, the gene with the strongest enrichment in SCZ relative to controls in SCHEMA, was not found to have a significantly higher burden in cases relative to controls in PGC3SEQ, suggesting that its effect size may have been overestimated in SCHEMA (ORPGC3SEQ = 1.6 vs. ORSCHEMA = 20.1).

I may suggest that the authors remove these indel calls from the analysis or perform experimental validation, while it should be recognized that such slippage can occur during PCR or even cell division. It might be also recommended to scrutinize other indel calls at polynucleotide stretches.

Also, as I commented, information on cDNA and protein changes caused by the variants in Table S8 and the IDs of corresponding transcripts would be useful to the readers.

Reviewer #3:

Remarks to the Author:

The authors have addressed my concerns. I have no further comments on this manuscript. It was a pleasure to read this well written paper.

Author Rebuttal, first revision:

Introductory Statement from the Authors: We thank the reviewers and editorial staff of *Nature Genetics* for the time and effort in further assessing our revision and giving constructive feedback. We are glad that two of the reviewers were content with our responses and revision. We also very

much appreciate the additional comments from reviewer #2 on the Indel calls, which prompted us to conduct further investigations that enhanced the rigor of the study and did not change any of the primary conclusions of the study. Specifically, these investigations showed that our primary conclusions held even if Indels, especially those particularly prone to sequencing errors in homopolymer regions, were excluded. Below, in blue text, we have provided point-by-point responses to the comments of the reviewers. Newly added texts to the main manuscript are highlighted in red.

Reviewers' Comments:

Reviewer #1:

Remarks to the Author:

The authors have addressed all the concerns properly.

We are glad that the review had found our responses and revision satisfying.

Reviewer #2:

Remarks to the Author:

The authors have adequately addressed most of the concerns I have raised, while I don't exactly agree with the rationale indicating that there is a certain level of a priori possibility that the overall effect of rare PTVs greatly varies across ethnicities (e.g., it is unsurprising that frequencies of individual rare PTVs derived from recent de novo mutations are different across populations and this fact does not directly support possible differences of the overall effect of rare PTVs across ethnicities; the example of MHC is for common polymorphisms and not for rare variants). Nevertheless, as I wrote, an analysis using real-world data from multiple ethnicities has a certain significance.

On the other hand, and perhaps more importantly, I have a specific concern on the accuracy of the indel calls below (one in case and four in controls) identified at a C stretch.

SETD1A 16 30977316 30965995 GCC GC frameshift variant NA 1 1 0

SETD1A 16 30977316 30965995 GCC G frameshift variant NA 1 0 1

SETD1A 16 30977321 30966000 CCC C frameshift variant NA 1 0 1

SETD1A 16 30977322 30966001 CC C frameshift variant NA 1 0 1

SETD1A 16 30977323 30966002 CG G frameshift variant NA 1 0 1

If these are false positives, it is not appropriate to conclude as follows:

Notably, SETD1A, the gene with the strongest enrichment in SCZ relative to controls in SCHEMA, was not found to have a significantly higher burden in cases relative to controls in

PGC3SEQ, suggesting that its effect size may have been overestimated in SCHEMA (ORPGC3SEQ = 1.6 vs. ORSCHEMA = 20.1).

I may suggest that the authors remove these indel calls from the analysis or perform experimental validation, while it should be recognized that such slippage can occur during PCR or even cell division. It might be also recommended to scrutinize other indel calls at polynucleotide stretches.

We whole-heartedly agree that validation of Indels, especially those involve homopolymers repeats of the same nucleotide, should be performed whenever possible (regardless of the sequencing technology, as these regions also remain a problem in Illumina data), and we share the concern with the reviewer. The lack of available additional aliquots of DNA for many samples with variants of interest prohibits such validation, and we now added this as a limitation in the manuscript. We were, however, able to show in below analyses that our primary findings concerning rare PTVs still held, even when Indels or homopolymer Indels were excluded.

We repeated the two main analyses in this study, the global enrichment and the gene-level burden test, with Indels overlapping any homopolymer stretches (25.7% of all tested Indels) excluded, and compared the new results with the original version. The comparisons are summarized and visualized in Table R1, Table R2 and Figure R1.

Table R1. Logistic regression for the global enrichment of rare variants

	OR	CI lower	CI upper	P-value
PTV	1.48	1.24	1.78	5.41E-06
PTV excluding homopolymer Indels	1.48	1.23	1.79	2.34E-05
PTV excluding all Indels	1.81	1.35	2.47	7.47E-05

Figure R1. Comparing gene-level burden p-values (right) and OR (left) for all PTVs (x axis) and for PTVs excluding Indels at homopolymer sites (y axis). Highlighted in blues are the nine genes implicated in SCHEMA and the three newly identified genes in PGC3SEQ.

Table 2. Gene-level burden test results for the twelve genes highlighted in Figure R1.

Gene	All PTVs				PTVs excluding homopolymer Indels			
	OR	p-value	count in cases	count in controls	OR	p-value	count in cases	count in controls
AKAP11	4.255	0.014	14	3	5.015	0.024	11	2
CACNA1G	0.421	0.105	6	13	0.280	0.026	4	13
CUL1	Inf	0.032	6	0	Inf	0.032	6	0
GRIN2A	0.000	0.477	0	1	0.000	0.477	0	1
HERC1	4.103	0.069	9	2	4.103	0.069	9	2
PCLO	5.015	0.024	11	2	Inf	0.008	8	0
RB1CC1	Inf	0.002	10	0	Inf	0.016	7	0
SETD1A	1.641	0.431	9	5	3.647	0.113	8	2
SP4	Inf	1.000	1	0	Inf	1.000	1	0

SRRM2	9.117	0.013	10	1	7.294	0.041	8	1
TRIO	0.911	1.000	3	3	0.911	1.000	3	3
XPO7	Inf	0.064	5	0	Inf	0.064	5	0

Table R1 shows that the global enrichment of PTV in SCZ cases as compared to controls was not affected by the exclusion of homopolymer Indels. When all Indels were excluded (50% of all rare PTVs), the enrichment indeed became stronger. Figure R1 and Table R2 demonstrate that individual genes had largely consistent effects when homopolymer Indels were included and when they were excluded. Specifically, among the twelve genes implicated in SCHEMA and/or PGC3SEQ, six of them had homopolymer Indels and the most notable change in gene's effect was observed for *SETD1A*. After removing homopolymer indels, the counts of rare PTV in cases/controls changed from 9/5 to 8/2, and the OR increased from 1.6. to 3.6. Both effect estimates were not statistically significant. Given its $OR_{SCHEMA} = 20.1$, the conclusion that *SETD1A* was likely overestimated in SCHEMA still holds.

Since Indels, regardless of whether a homopolymer is involved or not, constitute half of the rare PTVs, most of the genes would have too few counts to give meaningful effect estimates if all Indels are removed, and this practice would be inconsistent with practices followed in previous work (e.g., SCHEMA).

We believe this additional analysis will enhance the rigor of this study and will be of interest to readers who share the same concern, and have therefore added the section "Sensitivity analysis of Indels" to the Supplementary file, as well as a brief summary to the discussion stating the robustness of the finding and acknowledging the lack of Sanger validation as a limitation.

"The Ion Torrent technology is known to have reduced sequencing accuracy for Indels involving homopolymer repeats of the same nucleotide⁵³. We assessed the impact of Indel variants on our findings via a sensitivity analysis and demonstrated that excluding homopolymer Indels did not change our conclusion (Supplementary file)."

Also, as I commented, information on cDNA and protein changes caused by the variants in Table S8 and the IDs of corresponding transcripts would be useful to the readers.

Thank you for this suggestion. The updated Table S8 now contains information on the protein changes and the corresponding transcript of each variant.

Reviewer #3:

Remarks to the Author:

The authors have addressed my concerns. I have no further comments on this manuscript. It was a pleasure to read this well written paper.

Thank you.

Decision Letter, second revision:

16th September 2022

Dear Dongjing,

Thank you for submitting your revised manuscript "Schizophrenia risk conferred by rare protein-truncating variants is conserved across diverse human populations" (NG-LE58903R3). In light of your responses to Reviewer #2's concerns regarding potential miscalling of indels at homopolymer stretches and your additional sensitivity analyses showing that the primary conclusions hold when homopolymer indels are excluded from the analyses, we will be happy in principle to publish your study in Nature Genetics as a Letter pending final revisions to comply with our editorial and formatting guidelines.

We are now performing detailed checks on your paper and we will send you a checklist detailing our editorial and formatting requirements soon. Please do not upload the final materials and make any revisions until you receive this additional information from us.

Thank you again for your interest in Nature Genetics. Please do not hesitate to contact me if you have any questions.

Sincerely,
Kyle

Kyle Vogan, PhD
Senior Editor
Nature Genetics
<https://orcid.org/0000-0001-9565-9665>

Final Decision Letter:

23rd January 2023

Dear Dongjing,

I am delighted to say that your manuscript "Schizophrenia risk conferred by rare protein-truncating variants is conserved across diverse human populations" has been accepted for publication in an upcoming issue of Nature Genetics.

Over the next few weeks, your paper will be copyedited to ensure that it conforms to Nature Genetics style. Once your paper is typeset, you will receive an email with a link to choose the appropriate publishing options for your paper, and our Author Services team will be in touch regarding any additional information that may be required.

Your paper will be published online after we receive your corrections and will appear in print in the next available issue. You can find out your date of online publication by contacting the Nature Press Office (press@nature.com) after sending your e-proof corrections. Now is the time to inform your Public Relations or Press Office about your paper, as they might be interested in promoting its publication. This will allow them time to prepare an accurate and satisfactory press release. Include your manuscript tracking number (NG-LE58903R4) and the name of the journal, which they will need when they contact our Press Office.

Before your paper is published online, we will be distributing a press release to news organizations worldwide, which may very well include details of your work. We are happy for your institution or funding agency to prepare its own press release, but it must mention the embargo date and Nature Genetics. Our Press Office may contact you closer to the time of publication, but if you or your Press Office have any enquiries in the meantime, please contact press@nature.com.

Please note that Nature Genetics is a Transformative Journal (TJ). Authors may publish their research with us through the traditional subscription access route or make their paper immediately open access through payment of an article-processing charge (APC). Authors will not be required to make a final decision about access to their article until it has been accepted. [Find out more about Transformative Journals](https://www.springernature.com/gp/open-research/transformative-journals)

Authors may need to take specific actions to achieve > **compliance with funder and institutional open access mandates**. If your research is supported by a funder that requires immediate open access (e.g. according to [Plan S principles](https://www.springernature.com/gp/open-research/plan-s-compliance)) then you should select the gold OA route, and we will direct you to the compliant route where possible. For authors selecting the subscription publication route, the journal's standard licensing terms will need to be accepted, including <https://www.nature.com/nature-portfolio/editorial-policies/self-archiving-and-license-to-publish>. Those licensing terms will supersede any other terms that the author or any third party may assert apply to any version of the manuscript.

Please note that Nature Portfolio offers an immediate open access option only for papers that were first submitted after 1 January 2021.

An online order form for reprints of your paper is available at <https://www.nature.com/reprints/author-reprints.html>>. Please let your coauthors and your institutions' public affairs office know that they are also welcome to order reprints by this method.

If you have not already done so, we invite you to upload the step-by-step protocols used in this manuscript to the Protocols Exchange, part of our on-line web resource, natureprotocols.com. If you complete the upload by the time you receive your manuscript proofs, we can insert links in your article that lead directly to the protocol details. Your protocol will be made freely available upon publication of your paper. By participating in natureprotocols.com, you are enabling researchers to more readily reproduce or adapt the methodology you use. [Natureprotocols.com](https://natureprotocols.com) is fully searchable, providing your protocols and paper with increased utility and visibility. Please submit your protocol to <https://protocolexchange.researchsquare.com/>. After entering your nature.com username and password you will need to enter your manuscript number (NG-LE58903R4). Further information can be found at <https://www.nature.com/nature-portfolio/editorial-policies/reporting-standards#protocols>

Sincerely,
Kyle

Kyle Vogan, PhD
Senior Editor
Nature Genetics
<https://orcid.org/0000-0001-9565-9665>